# Bergamottin Inhibits Bovine Viral Diarrhea Virus Replication by Suppressing ROS-Mediated Endoplasmic Reticulum Stress and Apoptosis

**DOI:** 10.3390/v16081287

**Published:** 2024-08-13

**Authors:** Jinhua Yin, Jialu Zhang, Yi Liu, Cong Duan, Jiufeng Wang

**Affiliations:** 1College of Veterinary Medicine, China Agricultural University, Beijing 100193, China; yinjinhuadky@126.com (J.Y.); 18404969065@163.com (J.Z.); ly2990571963@163.com (Y.L.); 2College of Animal Science and Technology, Tarim University, Alar 843300, China; 3China Institute of Veterinary Drug Control, Beijing 100081, China

**Keywords:** bergamottin, BVDV, replication, apoptosis, endoplasmic reticulum stress, reactive oxygen species

## Abstract

Bovine viral diarrhea virus (BVDV) is one of the most important etiological agents that causes serious economic losses to the global livestock industry. Vaccines usually provide limited efficacy against BVDV due to the emergence of mutant strains. Therefore, developing novel strategies to combat BVDV infection is urgently needed. Bergamottin (Berg), a natural furanocoumarin compound, possesses various pharmaceutical bioactivities, but its effect on BVDV infection remains unknown. The present study aimed to investigate the antiviral effect and underlying mechanism of Berg against BVDV infection. The results showed that Berg exhibited an inhibitory effect on BVDV replication in MDBK cells by disrupting the viral replication and release, rather than directly inactivating virus particles. Mechanistically, Berg inhibits BVDV replication by suppressing endoplasmic reticulum (ER) stress-mediated apoptosis via reducing reactive oxygen species (ROS) generation. Studies in vivo demonstrated that oral gavage of Berg at doses of 50 mg/kg and 75 mg/kg significantly reduced the viral load within the intestines and spleen in BVDV-challenged mice. Furthermore, histopathological damage and oxidative stress caused by BVDV were also mitigated with Berg treatment. Our data indicated that Berg suppressed BVDV propagation both in vitro and in vivo, suggesting it as a promising antiviral option against BVDV.

## 1. Introduction

Bovine viral diarrhea virus (BVDV or *Pestivirus bovis*) is the causative pathogen of bovine viral diarrhea (BVD), recognized as one of the most economically significant cattle diseases in the world [1]. It is a positive-sense single-stranded RNA virus that belongs to the genus *Pestivirus* within the *Flaviviridae* family [2]. BVDV is further segregated into two biotypes, cytopathic (cp) and noncytopathic (ncp), according to its capacity to induce cytopathic effects (CPEs) in vitro. Since its initial discovery in 1946 in the United States, BVDV has been reported in numerous countries across the globe [3,4,5,6]. BVDV infection manifests various clinical symptoms, including diarrhea, abortion, reproductive disorders, and even a potentially lethal mucosal disease. These clinical manifestations contribute to a high mortality and mortality rate, resulting in potential economic losses estimated to be between USD 10 and 40 million per million calvings [7].

Currently, vaccination is one of the most commonly adopted measures to combat BVDV infection, whereas antivirals with established efficacy remain lacking [8]. However, existing vaccines may not always provide effective protection and are sometimes applied hesitantly due to genetic diversity and the emergence of mutants [9,10]. Therefore, it is urgent that we develop antiviral drugs with proven efficacy to prevent and control BVDV infection.

In recent years, the investigation of antiviral agents has attracted researchers’ attention. Some compounds have been tested as potential anti-BVDV agents, such as forsythiaside A [11], quercetin [12], matrine, and icariin [13]. The nucleoside analogs have high efficacy against BVDV infection in vitro, but there are negative side effects (immunosuppression, cytotoxicity) associated with them [14]. Thus, the development of novel strategies is urgent. Natural products derived from plants represent a valuable resource in the quest for antiviral agents due to their high biocompatibility and safety profile. Bergamottin, (Berg), also known as 5-geranoxypsoralen, is a natural furanocoumarin compound commonly found in fruits of the *Citrus* family such as lemon oil and grapefruit juice [15]. It exhibits various pharmaceutical bioactivities, including anti-inflammatory, antioxidant, and anticancer properties [16,17]. In recent years, several studies have highlighted Berg’s antiviral effects against diverse types of viruses. For instance, it has been demonstrated to interfere with the entry and replication steps of SARS-CoV-2, specifically by reducing ACE2 protein expression [18,19]. Another study showed Berg’s inhibitory effect on Lassa virus (LASV) by blocking viral entry and endocytic trafficking [20]. Additionally, Berg possesses antiviral activity against authentic lymphocytic choriomeningitis virus (LCMV) infection [20]. However, whether Berg exhibits inhibitory effect on BVDV infection remains unknown.

The endoplasmic reticulum (ER) is a crucial cellular membrane organelle in eukaryotic cells. Viruses naturally employ the host’s translation machinery to produce an extensive amount of viral proteins in the ER lumen, which will disturb the ER homeostasis and ultimately result in unavoidable ER stress [21]. To cope with the deleterious effects of ER stress, the unfolded protein response (UPR) is triggered to restore ER homeostasis [22]. Mounting evidence suggests that ER stress and persistent UPR play a pivotal role in the pathogenesis of viral infection disease, especially ER-tropic viruses such as the *Flaviviridae* family [23,24,25]. In the case of BVDV infection, the cellular ER stress and UPR pathway are activated to maintain a favorable environment for its propagation [26,27]. Previous research has demonstrated that Berg could mitigate ER stress in cafeteria-diet-fed mice [28]. Nevertheless, it remains unclear whether Berg affects ER stress induced by BVDV.

Mounting evidence indicates that persistent ER stress and prolonged UPR can trigger apoptosis. Apoptosis serves as an innate host defense mechanism aiming at eradicating pathogen-infected cells. However, viruses have evolved various strategies to manipulate apoptosis to promote their reproduction and spread [29]. Previous reports have demonstrated that cp BVDV-infected cells often exhibit CPE and display typical apoptosis features [30,31,32,33]. Notably, apoptosis induced by cp BVDV is directly linked to oxidative stress [34] and ER stress [35,36]. Evidence has emerged that cp BVDV induces ER stress-mediated apoptosis in MDBK and bovine placental trophoblast cells (BTCs) to facilitate its propagation [26,35]. The suppression of ER stress-mediated apoptosis dramatically reduces both BVDV multiplication and pathogenicity. Therefore, further exploration is required to elucidate the correlation between Berg’s antiviral activity and its impact on ER stress, oxidative stress, and apoptosis.

In this study, we explored the antiviral activity and the underlying mechanism of Berg against BVDV infection. Our results indicate that Berg inhibited BVDV proliferation both in vitro and in vivo in a dose-dependent manner. Mechanistic studies revealed that Berg disrupted the BVDV replication and release stage, rather than directly inactivating virus particles. Additionally, we found that Berg suppressed ER stress-mediated apoptosis induced by BVDV via reducing ROS production. Overall, our findings suggest that Berg has excellent potential as an anti-BVDV agent.

## 2. Materials and Methods

### 2.1. Cell Lines, Viruses, and Reagents

Madin–Darby bovine kidney (MDBK) cells were procured from the China Veterinary Culture Collection Center (Beijing, China) and maintained in Dulbecco’s modified Eagle’s medium/Ham’s F-12 medium (DMEM/F-12, Gibco, Grand Island, NY, USA) containing 1% penicillin–streptomycin and 10% (vol/vol) fetal bovine serum (FBS, Gibco, Grand Island, NY, USA). BVDV 1-NADL strain was stored in our laboratory. The MDBK cells and FBS in this study were free of adventitial ncp BVDV.

Berg (purity > 99%) and Ribavirin (Rib) were obtained from MedChemExpress (Shanghai, China). Drug stocks were dissolved in dimethyl sulfoxide (DMSO, Sigma-Aldrich, Saint Louis, MO, USA) and diluted to a concentration of 1 mM with DMEM/F-12 medium for storage at −80 °C. The content of DMSO in the culture medium was below 0.1%.

### 2.2. Cytotoxicity and Antiviral Activity Assay

The viability of MDBK cells in the presence and absence of Berg was assessed using the Cell Counting Kit-8 (CA1210, Solarbio, Beijing, China). Briefly, MDBK cells were incubated with increasing concentrations of Berg. After 24, 48, and 72 h, the CCK-8 solution was applied to each well and the cells were incubated for 2 h at 37 °C. The absorbance of the plate was measured at a wavelength of 450 nm using a microplate reader. The cell viability was calculated according to the following formula: cell viability (%) = [A (experimental group) − A (blank group)]/[A (control group) − A (blank group)] × 100. To assess the antiviral efficacy of Berg, cells were infected with BVDV for 1 h at MOI = 1. Subsequently, cells were treated with ribavirin and different concentrations of Berg (25, 50, and 100 μM) at 24, 48, and 72 h post infection (hpi). In each plate, negative controls (DMEM/F-12) and infected positive controls (BVDV alone) were included. RT-qPCR, Western blotting, and IFAs were applied to quantify viral replication.

### 2.3. Real-Time Quantitative Reverse-Transcription PCR (RT-qPCR)

Total RNA from samples was extracted using an RNAiso Plus kit (Takara, SanJose, CA, USA). Then, the extracted RNA was used to synthesize cDNA with the cDNA Synthesis Kit (Vazyme Biotechnology, Nanjing, China). The specific primers were as follows: BVDV 5′UTR (Forward: 5′-TAGTCGTCAGTGGTTCGACGCC-3′; Reverse: 5′-CCTCTGCAGCACCCTATCAG-3′) and GAPDH (Forward: 5′-AAAGTGGACATCGTCGCCAT-3′; Reverse: 5′-CCGTTCTCTGCCTTGACTGT-3′). The RT-qPCR assay was conducted on a 7500 Real-Time PCR System (Applied Biosystems) using a qPCR Kit (AUQ-01, TranGen Biotechnology, Beijing, China). The RT-qPCR procedure was as follows: 95 °C for 30 s, and 45 amplification cycles in two steps of 95 °C for 5 s and 60 °C for 30 s. Then, the melting curve was carried out as follows: 95 °C for 15 s, 60 °C for 60 s, 95 °C for 15 s, and 60 °C for 15 s. GAPDH was used as the reference gene. The threshold cycle (ΔCt) value was calculated as ΔCt = Ct(5′UTR) − Ct(GAPDH). The relative gene expression levels were calculated by the 2^−ΔΔCt^ method. The absolute RNA quantities of supernatants were detected as previously described [37].

### 2.4. Virus Titration

The 50% tissue culture infective dose (TCID_50_) was used to determine the virus titer of BVDV. Briefly, MDBK cells were seeded in 96-well plates for 20 h at 37 °C and then infected with 10-fold dilutions (10^−1^ to 10^−10^) of BVDV samples in five replicates for 2 h at 37 °C. Next, the inoculum was removed and replaced with the cell culture medium (containing 2% FBS) and the plates were incubated at 37 °C for a further 48 h. After that, the viral proteins were viewed via the IFA. The viral titer was calculated using the Reed–Muench method. The titer values were recorded as log_10_ (TCID_50_) per milliliter.

### 2.5. Indirect Immunofluorescence Assay (IFA)

Cells were seeded onto coverslips in 24-well cell culture plates and infected with BVDV for 1 h prior to Berg treatment. After 24 hpi, the cells were fixed with 4% paraformaldehyde for 20 min. Subsequently, the cells were permeabilized using 1% Triton X-100 solution for 15 min, then blocked with 2% bovine serum albumin (BSA) for 1 h at room temperature. The cells were further incubated with a rabbit anti-E2 polyclonal antibody (1:500, prepared in our laboratory) overnight at 4 °C. Following washes with PBS three times, the cells were incubated with goat anti-rabbit secondary antibody labeled with Alexa Fluor 488 (Beyotime Biotechnology, Shanghai, China) for 1 h at 37 °C. Finally, the cells were incubated with 4′-6-diamidino-2-phenylindole (DAPI, Solarbio, Beijing, China) for 6 min. Thereafter, images of the staining samples were acquired using LAS X 4.7.0 software and a confocal laser scanning microscope (Leica SP8, Wetzlar, Germany).

### 2.6. Western Blotting

Cells were harvested and lysed using radioimmunoprecipitation assay (RIPA) buffer supplemented with 1% phenylmethanesulfonyl fluoride (PMSF) for 30 min at 4 °C. The lysates were boiled for 10 min in 5 × SDS-PAGE loading buffer. Subsequently, the proteins were subjected to sodium dodecyl sulfate polyacrylamide gel electrophoresis (SDS-PAGE) at the voltage of 80 V. The proteins from gels were transferred onto a polyvinylidene difluoride (PVDF) membrane. The membranes were blocked with a 2% BSA for 1 h to minimize background and incubated with the following primary antibodies overnight at 4 °C: GRP78 (1:1000) from Wanlei Bio (Shenyang, China), phosphor-PERK (1:1000), phospho-eIF2α (1:1000), eIF2α (1:5000), ATF-4 (1:1000), CHOP (1:1000), Bax (1:1000), Bcl-2 (1:1000), Caspase 3 (1:1000), Cleaved caspase 3 (1:1000), α-tubulin (1:20,000), and β-actin (1:20,000) from Cell Signaling Technology (Danvers, MA, USA). After washing, the membranes were incubated with HRP-conjugated goat anti-rabbit IgG or anti-mouse IgG from Proteintech Group, Inc. (Rosemont, IL, USA) for 1 h at 37 °C. Images of target protein bands were captured using a Tanon 6200 chemiluminescence imaging workstation and analyzed using Image J 1.8.0 software.

### 2.7. Virus Inactivation and Replication Cycle Assay

To examine whether Berg has a direct effect on the virus, BVDV (MOI = 1) was mixed with different concentrations of Berg (25, 50, or 100 μM) and incubated for 1 h at 37 °C. Subsequently, the solution underwent centrifugation at 90,000× *g* for 1.5 h at 4 °C to separate the virus particles. The isolated viruses were then suspended in cell culture medium and infected MDBK cells for 24 h. Afterwards, total RNA was extracted for RT-qPCR and the cells were fixed for IFA to detect the virus infectivity.

For the viral attachment assay, MDBK cells were pretreated with Berg (50 or 100 μM) for 1 h, followed by infection with BVDV (MOI = 1) for 2 h at 4 °C. Cells were collected after washing with ice-cold PBS to measure viral RNA levels by RT-qPCR. For the viral internalization assay, MDBK cells were infected with BVDV (MOI = 1) at 4 °C for 1 h and washed with PBS (pH 3.0) to remove non-internalized virus. Then, the cells were treated with Berg (50 or 100 μM) for 1 h at 37 °C and collected to measure viral RNA levels by RT-qPCR. For the viral replication assay, MDBK cells were infected with BVDV (MOI = 1) for 1 h at 37 °C and washed with PBS (pH 3.0) to remove the extracellular virus. Following that, cells were inoculated with Berg at the final concentration of 50 or 100 μM for 10 h at 37 °C and collected to measure viral RNA levels by RT-qPCR. For the viral release assay, MDBK cells were infected with BVDV (MOI = 1) for 1 h at 37 °C. Unbound viruses were removed by PBS, and the culture medium was immediately changed with fresh DMEM/F-12 medium. At 10 hpi, different concentrations of Berg (50 or 100 μM) were added to the cells for another 2 h at 37 °C. Supernatants were harvested to analyze viral RNA copies by RT-qPCR.

### 2.8. Imaging of ER Morphology

The uninfected or BVDV-infected MDBK cells were incubated with the various concentrations of Berg (50 or 100 μM) for 24 h. Then, the cells were incubated with ER-tracker red working solution at 37 °C for 30 min. Following this, the cells were stained with Hoechst 33342 dye for 6 min. The images of ER morphology were viewed via a confocal laser scanning microscope (Leica SP8, Wetzlar, Germany).

### 2.9. TUNEL Staining

The uninfected or BVDV-infected MDBK cells were treated with various concentrations of Berg (50 or 100 μM) Berg for 24 h, followed by the TUNEL assay, performed following the manufacturer’s protocol (Vazyme Biotechnology, Nanjing, China). In brief, MDBK cells were fixed with 4% paraformaldehyde for 1 h, permeabilized for 15 min using 1% Triton X-100, and then treated with TUNEL working solution for 1 h. Afterwards, DAPI dye was added to stain cell nuclei for 6 min. The images were acquired using a confocal laser scanning microscope (Leica SP8, Wetzlar, Germany).

### 2.10. Determination of Mitochondrial Membrane Potential (∆Ψm)

To assess the onset of apoptosis, mitochondrial membrane potential (∆*Ψm*) was measured using the mitochondrial membrane potential assay kit with JC-1 (Beyotime Biotechnology, Shanghai, China). MDBK cells were infected with BVDV (MOI = 1) and treated with various concentrations of Berg (25, 50, or 100 μM) for 24 h. After incubation with JC-1 working solution at 37 °C for 20 min, the medium was replaced with DMEM/F-12. The fluorescence signals were photographed using LAS X 4.7.0 software and a confocal laser scanning microscope (Leica SP8, Wetzlar, Germany).

### 2.11. Detection of the Antioxidant Status in Cells and Tissues

The levels of superoxide dismutase (SOD), glutathione (GSH), and malondialdehyde (MDA) and the total antioxidant capacity (T-AOC) in MDBK cells and serum samples from mice were detected using commercially available test kits (S0101S and S0059S, Beyotime, Beijing, China; A003-1-2 and A015-2-1, Nanjing Jiancheng Bioengineering Institute, Nanjing, China) following the instructions provided by the manufacturer.

### 2.12. Measurement of Intracellular Reactive Oxygen Species (ROS) Levels

The uninfected or BVDV-infected MDBK cells were incubated with various concentrations of Berg (50 or 100 μM) for 24 h. Then, the cells were washed with serum-free DMEM/F-12 and incubated with 10 μM DCFH-DA at 37 °C for 20 min. After washing with DMEM/F-12 three times, fluorescence signals were captured by a fluorescence microscope. The fluorescence intensity was quantified using Image J software.

### 2.13. Animal Experiments

Female BALB/c mice (6–8 weeks old) from SPF Biotechnology Co., Ltd. (Beijing, China) were raised in the laboratory animal facility for animal care and use. All animal experiments were carried out in compliance with the experimental protocol approved by the Committee of Animal Use and Protection of China Agricultural University (AW92404202-2-1).

All mice were randomly divided into 5 groups (n = 8): control, BVDV, BVDV + Berg (50 mg/kg), BVDV + Berg (75 mg/kg), and Berg (75 mg/kg) (dosage setting referenced from previous studies [14]). In the BVDV infection group, each mouse received a challenge of 6 × 10^6^ TCID_50_ virus via intraperitoneal (i.p.) route at 0 dpi, following the protocol established in our prior research [30]. In the Berg protection groups, BVDV-challenged mice were orally administrated Berg (dissolved with 0.5% DMSO, at either 50 or 75 mg/kg body weight per mouse) daily for 6 consecutive days. In the control and Berg groups, mice were treated with an equal volume of PBS containing 0.5% DMSO and Berg (75 mg/kg) once a day for 6 consecutive days, respectively. Throughout the experiment, the body weights and temperatures of mice were monitored daily. At the end of the experiments, sera were collected to assess the levels of T-AOC, MDA, and SOD. Furthermore, the lungs, spleen, and intestine were harvested from each mouse for investigating viral propagation and conducting histopathological analysis in the BVDV-challenged mice. Pathological lesions in the relevant tissues were assessed and scored based on referencing our previous study [38].

### 2.14. Immunohistochemistry (IHC)

The viral antigens in the spleen were examined by IHC. Briefly, spleen sections were dewaxed and antigens were retrieved with citrate buffer (pH 6.0). Afterward, the endogenous peroxidase was blocked by 3% hydrogen peroxide for 10 min at room temperature. Subsequently, the sections were blocked with 10% goat serum for 30 min at room temperature and incubated with rabbit anti-BVDV E2 polyclonal antibody (prepared in our laboratory) overnight at 4 °C. Then, the HRP-conjugated goat anti-rabbit IgG was added to the slides and incubated for 20 min at 37 °C. Following color development using 3,3′-diaminobenzidine (DAB, Zhongshan Golden Bridge Biotechnology, Beijing, China), images of sections were viewed under a light microscope.

### 2.15. Statistical Analysis

All data are presented as the mean ± standard deviation (SD) from three independent experiments. GraphPad Prism 8.0 software (GraphPad Software, San Diego, CA, USA) was utilized to perform statistical analysis. The statistical significance of differences was assessed by one-way ANOVA (*** *p* < 0.001, ** *p* < 0.01, * *p* < 0.05).

## 3. Results

### 3.1. Berg Displays Antiviral Activity of BVDV in MDBK Cells

The chemical structure of Berg is depicted in Figure 1A. The cell viability results showed that Berg exhibited no significant cytotoxicity up to a concentration of 100 µM at 24, 48, and 72 h (Figure 1B). Consequently, concentrations of 25, 50, and 100 µM Berg were set for subsequent investigations. Further characterization of its effect on BVDV propagation was conducted by treatment with increasing concentrations of Berg. As shown in Figure 1C–F, Berg treatment dramatically decreased the viral titers and RNA levels in a dose-dependent manner at 24, 48, and 72 h. Notably, Berg maintained potent antiviral activity even at 72 hpi. Western blotting analysis revealed a significant reduction in BVDV E2 protein expression in the Berg treatment group (Figure 1G). In line with these findings, a clear dose-dependent reduction in fluorescence intensity of BVDV E2 was observed upon Berg treatment (Figure 1H), indicating the inhibition of BVDV E2 synthesis by Berg.

As ribavirin is a broad-spectrum antiviral drug, it was used as the positive control in this study. The results presented in Figure 1C–G indicate that ribavirin treatment remarkably reduced intracellular viral RNA, titers, and protein levels. Notably, the antiviral efficacy of Berg in vitro appeared comparable to ribavirin. Therefore, these findings suggest that Berg effectively inhibits BVDV replication without causing measurable cytotoxicity.

### 3.2. Berg Blocks the Replication and Release of BVDV, Rather Than Inactivating Virus Particles

To elucidate the mode of action of Berg in inhibiting BVDV replication, an inactivating assay was conducted. Different concentrations of Berg were incubated with BVDV, after which the virus was separated to infected MDBK cells. Finally, cell samples were harvested to measure the viral infectivity by RT-qPCR and IFAs at 24 hpi (Figure 2A). As shown in Figure 2B, no significant difference in viral RNA levels was observed even at a concentration of 100 μM Berg treatment. Similarly, there was no noticeable reduction in the fluorescence intensity of BVDV E2 protein upon Berg treatment (Figure 2C), demonstrating that Berg does not exert viricidal activity on BVDV. To further investigate whether Berg’s inhibitory effect on BVDV replication is equally effective at different stages of the viral life cycle, Berg was added to the BVDV-infected cells at the indicated time points. As shown in Figure 2D,E, no significant decrease in RNA levels was observed upon Berg treatment, indicating that Berg did not affect viral attachment and internalization. Subsequently, the effect of Berg on BVDV replication and release was evaluated. The results determined that Berg treatment markedly reduced viral RNA levels during both the virus replication and release steps (Figure 2F,G). Taken as a whole, these findings suggest that Berg disrupts the replication and release steps of BVDV, rather than directly inactivating virions.

### 3.3. Berg Attenuates BVDV-Induced ER Stress

BVDV hijacks ER as a replication platform during its life cycle and leads to ER stress, which is beneficial for its propagation. To explore whether Berg exhibits inhibitory effect on BVDV-triggered ER stress, ER tracker staining was conducted to visualize ER morphology. As shown in Figure 3B, ER swelling and increased fluorescence intensity in the ER tracker were evident in BVDV-infected cells, whereas Berg treatment reversed this phenomenon. Western blotting analysis revealed that the expression level of GRP78, a crucial marker protein of ER stress, was significantly enhanced in the BVDV group. Following treatment with Berg, the GRP78 protein decreased (Figure 3A). Moreover, Berg effectively inhibited the activation of p-PERK, p-eIF2α, ATF4, and CHOP induced by BVDV infection (Figure 3C). These results indicate that Berg suppresses the activation of ER stress and the PERK pathway in BVDV-infected cells.

### 3.4. Berg Inhibits BVDV-Induced Apoptosis

TUNEL and Western blotting assays were conducted to investigate whether Berg affected BVDV-induced apoptosis. As indicated in Figure 4A, the upregulation of Bax and cleaved caspase 3 protein expression, along with the downregulation of anti-apoptotic protein Bcl-2 expression, were observed in the BVDV group, indicating the activation of the apoptosis signaling pathway by BVDV. However, Berg treatment markedly inhibited Bax expression and the cleavage of caspase 3 protein but enhanced Bcl-2 expression. These findings were further corroborated by the TUNEL assay. Figure 4B illustrates a notable increase in TUNEL-positive cells in the BVDV-infected group compared to the uninfected group. Administration with various concentrations of Berg in BVDV-infected cells led to a notable decrease in the number of TUNEL-positive cells. Furthermore, the ∆*Ψm* assay was further applied to detect Berg’s effect on apoptosis induced by BVDV. Mitochondrial membrane depolarization was indicated by the transition of JC-1 from aggregate (red fluorescence) to monomer (green fluorescence). An increase in the ratio of green/red fluorescence signifies mitochondrial depolarization. As illustrated in Figure 4C,D, the ration of green to red fluorescence was markedly increased in BVDV-infected cells, indicating that BVDV infection decreased the ∆*Ψm* and resulted in mitochondrial dysfunction. However, this phenomenon was reversed by Berg treatment. These results suggest that Berg has the potential to prevent BVDV-induced apoptosis and mitochondrial depolarization.

### 3.5. Berg Suppresses ER Stress-Mediated Apoptosis in BVDV-Infected Cells

To explore the role of ER stress and apoptosis in Berg’s inhibition of BVDV replication, thapsigargin (TG), an ER stress activator, was used. As illustrated in Figure 5, the GRP78, CHOP, and cleaved caspase 3 proteins were significantly increased in the BVDV, TG, and BVDV + TG groups compared to the mock-treated (uninfected) group, indicating the activation of ER-stress and apoptosis in these groups. Furthermore, Berg treatment downregulated GRP78, CHOP, and cleaved caspase 3 protein expression, concomitant with a reduction in BVDV E2 protein levels. Of note, the application of TG reversed the suppressive effect of Berg on BVDV proliferation and the related protein expression involved in ER stress and apoptosis. This was evidenced by the increased protein levels of BVDV E2, GRP78, CHOP, and cleaved caspase 3 in the BVDV + TG + Berg group compared to the BVDV + Berg group. These results indicate that Berg inhibits BVDV replication, probably through the regulation of the ER stress-mediated apoptosis pathway.

### 3.6. Berg Inhibits BVDV Propagation through Suppression of ER Stress-Mediated Apoptosis via Reduction in ROS Generation

To detect the potential effect of Berg on BVDV-induced oxidative stress, ROS, GSH, and SOD levels were determined. The results showed that BVDV infection dramatically increased the intracellular ROS levels, while those were markedly reduced upon Berg treatment, as well as NAC administration (Figure 6A,B). Similarly, Berg treatment reversed the reduction in GSH and SOD, along with the increase in MDA levels, induced by BVDV infection (Figure 6C–E). The above results indicate that Berg alleviates BVDV-induced oxidative stress. Given that both ROS and ER stress could induce apoptosis in BVDV-infected cells, the roles of ROS, ER stress, and apoptosis in Berg’s inhibitory effect on BVDV proliferation were elucidated using 4-phenylbutyric acid (4-PBA, ER stress inhibitor) and N-acetylcysteine (NAC, an ROS scavenger). As shown in Figure 6F, the expression of Bax, cleaved caspase 3, and BVDV E2 was markedly upregulated, while Bcl-2 expression was downregulated following NAC and 4-PBA treatment. These results suggest that ROS and ER stress may contribute to Berg’s suppression of BVDV-induced apoptosis, thereby inhibiting BVDV replication. Additionally, the application of NAC in BVDV-infected cells remarkably decreased GRP78 protein expression. Collectively, these findings reveal that Berg inhibits BVDV replication through the suppression of ER stress-mediated apoptosis by reducing ROS generation.

### 3.7. Berg Exerts Antiviral Efficacy against BVDV Infection In Vivo

As depicted in the schematic diagram in Figure 7A, mice were orally administrated Berg (50 or 75 mg/kg) once daily for 6 days following BVDV challenge via i.p. injection. During this period, no significant changes were observed in the body weights and temperatures of mice (Figure 7B,C). As shown in Figure 7D–G, the viral RNA levels in the duodenum, jejunum, ileum, and spleen were reduced in all Berg-treated groups. Consistent with the viral RNA levels in tissues, Berg treatment also significantly decreased BVDV E2 expression in the spleen in a dose-dependent manner (Figure 7H,I). These findings indicate that Berg significantly inhibits BVDV propagation in vivo. Histology and eosinophil staining results indicated that BVDV infection induced the apparent pathological damages. In the intestinal tissue of BVDV-challenged mice, evident changes included intestinal villi tip breakage, inflammatory cell infiltration, and basement membrane mucosal edema. Lung tissue changes were characterized by the thickening of the alveolar wall and liver tissue exhibited disordered arrangement in the liver cells, inflammatory cell infiltration, and abnormal liver structure in BVDV-challenged mice. Additionally, spleen tissue from BVDV-challenged mice showed increased macrophages and a damaged spleen structure. Notably, Berg treatment markedly ameliorated these pathological damages. Moreover, tissues from mice administrated with Berg alone showed no significant changes compared to the control group, indicating that a dose of 75 mg/kg of Berg exhibited no measurable toxicity (Figure 7J). The histopathological scores are illustrated in Figure 7K. These results indicate that Berg treatment effectively reduces viral load and ameliorates the pathological damages in BVDV-challenged mice.

### 3.8. Berg Weakens ER Stress-Mediated Apoptosis through Antioxidant System in BVDV-Challenged Mice

To test the antiviral mechanism of Berg against BVDV infection in vivo, antioxidant enzymes in serum and protein levels in duodenum and spleen tissues were measured. As indicated in Figure 8A–C, notable decreases in SOD activity and T-AOC content, concomitant with an increase in MDA content were observed in BVDV-challenged mice. However, Berg treatment reversed these trends (Figure 8A–C). Moreover, Western blotting analysis showed a significant increase in GRP78, CHOP, Bax, and cleaved caspase 3 expression and a decrease in Bcl-2 expression in the duodenum and spleen in the BVDV-challenged group compared with the control group. Conversely, these effects were markedly reversed by the different dosages of Berg administration (Figure 8D). These results suggest that Berg treatment weakens ER stress-mediated apoptosis through the antioxidant system in BVDV-challenged mice.

## 4. Discussion

Bovine viral diarrhea virus (BVDV) poses a substantial threat to the global livestock industry, leading to considerable financial losses. Currently, vaccination is the main strategy to prevent BVDV infection. Nevertheless, the multiple subtypes and high rate of mutant strains lead to the efficacy of vaccines being limited. Therefore, it is necessary to develop novel drugs to control BVDV infection. Berg, a furanocoumarin compound, was reported to possess antiviral activity against a diverse array of viruses, such as SARS-CoV-2, LASY, and so on [18,19,20]. However, the antiviral mechanism in Berg remains unclear. In this study, we investigated the inhibitory effect of Berg on BVDV infection both in vitro and in vivo and further clarified the underlying mechanism of this action.

Our results demonstrated that Berg exhibited potent inhibition of BVDV propagation in MDBK cells in a dose-dependent manner. Importantly, Berg did not exert inactivating effect on BVDV, indicating that its anti-BVDV activity did not involve direct interaction with virus particles. The BVDV replication cycle includes attachment, internalization, replication, and release [39]. Each stage may be a potential target for antiviral therapy. Interestingly, Berg was found to prevent BVDV infection at the replication and release stage, implying that Berg’s predominant action occurred following viral entry. Previous research has shown that cell-specific receptors and clathrin-mediated endocytosis facilitate BVDV attachment and entry [40,41,42]. Therefore, it is plausible to speculate that Berg did not impede viral adhesion or entry by decreasing the expression of host–cell receptors such as CD 46, claudin-1, and occluding [41]. Given the evidence that Berg inhibited infectious BVDV production, a mouse model of BVDV infection was established to evaluate the antiviral efficacy in vivo. The present data show that Berg reduced viral load in target organs and mitigated BVDV-induced tissue pathological damages. Notably, a dose of 75 mg/kg exhibited superior therapeutic effectiveness compared to 50 mg/kg. These findings indicate the effectiveness of Berg in suppressing BVDV propagation both in vitro and in vivo, laying the groundwork for potential clinical applications.

ER stress is critical in the viral life cycle. In mammalian cells, the UPR is mediated by three distinct downstream signaling pathways (PERK, IRE1, and ATF6). Viruses interact with one or more branches of UPR to attenuate ER stress [43]. Our previous study revealed that BVDV enlarged the volume of ER and selectively activated PERK branch to cope with the ER stress, thereby contributing to viral proliferation and pathogenicity [26,27,35]. In this work, Berg alleviated the phenomenon of ER swelling and dramatically downregulated the expression of GRP78, p-PERK, p-eIF2α, ATF-4, and CHOP in BVDV-infected cells, suggesting that Berg suppressed the activation of ER stress and UPR following BVDV infection. Moreover, the ER stress inhibitor 4-PBA significantly suppressed BVDV proliferation, indicating that attenuation of the ER stress could prevent BVDV infection to some extent. These findings align with previous reports demonstrating Berg’s ability to mitigate liver damage via the attenuation of ER stress [28].

Given the pivotal role of ER stress in virus infection, targeting ER stress-mediated signaling pathways has emerged as a promising strategy for developing antiviral agents. It has been revealed that excessive and sustained ER stress can lead to apoptosis, which is essential in the context of BVDV infection [26,27,35]. Our finding showed that Berg reduced the expression of the key ER stress-mediated apoptosis regulator CHOP in BVDV-infected cells. Importantly, ER stress inducer TG promoted BVDV propagation and abolished the inhibitory effect of Berg on viral replication, which was in line with our previous published reports [27]. Also, TG upregulated CHOP and pro-apoptotic protein expression, indicating that ER stress-mediated apoptosis is involved in Berg’s antiviral effect against BVDV infection. These results were further verified in vivo.

Apoptosis acts as a crucial innate defense mechanism, aiming to remove intracellular pathogens and prevent them from multiplying and spreading. However, viruses have evolved various strategies to evade, postpone, or divert apoptosis to favor their replication [29]. Viruses typically delay or inhibit apoptosis during the early stage of viral cycle to ensure the completion of their replication cycle and maximize viral progeny production. Moreover, apoptosis can be triggered in the late stages of viral replication to facilitate the release of progeny virions and expedite the infection of neighboring cells [44,45]. Indeed, apoptosis has been implicated in facilitating viral release [46,47], and the use of apoptosis inhibitors has been shown to impede viral release [48]. Increasing evidence has shown that multiple viruses utilize apoptotic mimicry to facilitate their entrance and spread [49,50]. Moreover, some viruses have evolved to hijack apoptotic bodies at the late stage of infection for cell–cell transmission to evade host antiviral responses [45,51]. A recent study has documented that BVDV activates ER stress-mediated apoptosis during late infection for virus proliferation and spread [35]. In this study, we observed that Berg could inhibit cp BVDV-induced apoptosis, suggesting a potential association between this action and the suppression of viral release.

Alterations in redox homeostasis, such as changes in intracellular ROS levels, play a crucial role in promoting viral propagation and contributing to virus-induced diseases. Several papers have reported that BVDV infection induces oxidative stress [26,34,35,52,53]. Consistent with these reports, our study demonstrated that BVDV infection led to elevated ROS generation and decreased GSH and SOD levels both in vitro and in a virus-challenged mouse model. Nevertheless, treatment with Berg effectively alleviated BVDV-induced oxidant stress by decreasing MDA and increasing GSH and SOD levels. These findings provide evidence that Berg possesses antioxidant capacity, which is in line with prior studies [17,28]. Moreover, there is a significant interplay between ROS and ER stress in the context of cell apoptosis. Mitochondria, as the major source of ROS, play a pivotal role in cellular energy production [54]. ROS generated by mitochondria disrupt ER homeostasis, leading to ER stress. Conversely, ER stress can further increase mitochondria ROS via the release of Ca^2+^, thereby establishing a vicious circle that disrupts cellular homeostasis and triggers apoptosis [55,56]. Our results suggest that ROS may act upstream of the UPR, as evidenced by the significant inhibition of ER stress observed with NAC treatment. Furthermore, the observed increase in SOD levels in the Berg treatment highlights the importance of this antioxidant enzyme in protecting cells from oxidative stress. Similarly, GSH plays a crucial role in maintaining redox homeostasis and exhibits antiviral effect through various mechanisms, such as modulating the viral life cycle, regulating immunomodulatory activity, and interfering with protein folding and maturation [57]. Furthermore, GSH and analogs have been shown to be effective viral inhibitors [57]. Notably, our study demonstrated that Berg effectively preserved GSH levels in BVDV-infected cells, thereby alleviating oxidative stress and enhancing cellular antioxidant capacity. Overall, our findings demonstrate that Berg inhibits BVDV replication by suppressing the ER stress-mediated apoptosis via reducing ROS generation (Figure 9). These insights contribute to the development of potential antiviral drugs and provide a scientific basis for Berg as a potential anti-BVDV drug.

## 5. Conclusions

In conclusion, our study provides compelling evidence that Berg exhibits potent inhibitory effects on BVDV replication both in vitro and in vivo. In addition, the replication and release stage of the BVDV life cycle were suppressed by Berg. The mechanism of this action appears to be mediated by alleviating ER stress-mediated apoptosis via reducing ROS generation.

## Figures and Tables

**Figure 1 viruses-16-01287-f001:**
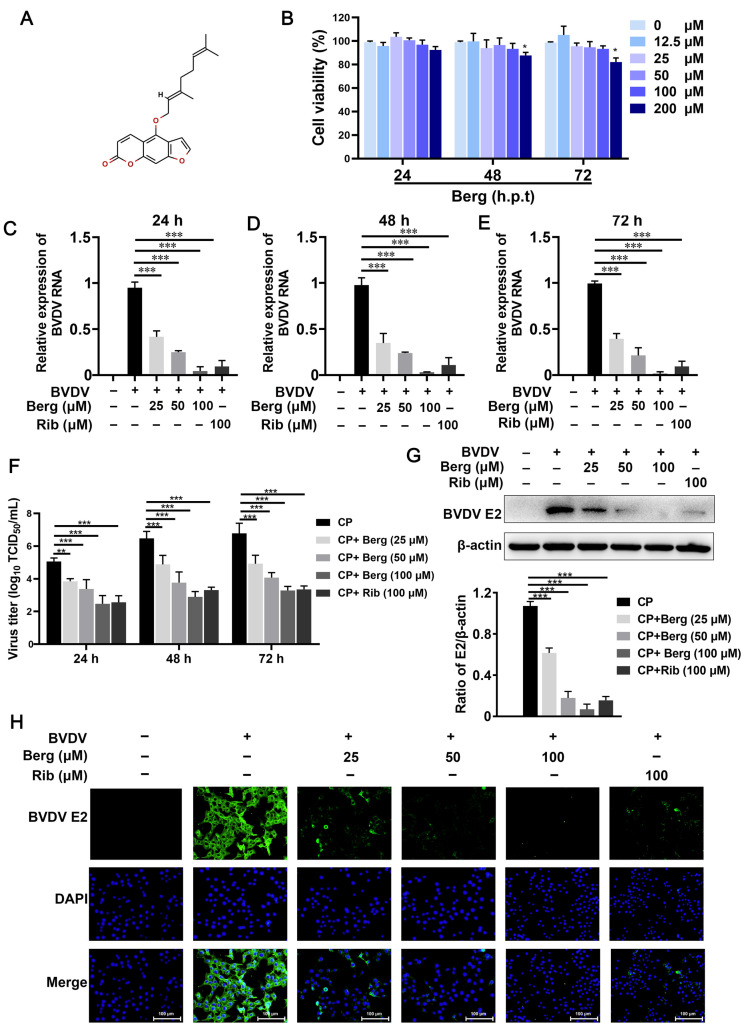
Cytotoxicity and anti-BVDV effect of Berg in MDBK cells. (**A**) Chemical structure of Bergamottin (Berg). (**B**) The cell viability of MDBK cells upon treatment with increasing concentrations of Berg was assessed by CCK-8 assay. (**C**–**F**) Viral RNA levels and titers in the BVDV-infected cells upon treatment with or without Berg (25, 50, and 100 μM) were quantified by RT-qPCR and TCID_50_ assay at 24, 48, and 72 hpi, respectively. BVDV E2 protein levels in the uninfected or BVDV-infected cells in response to indicated concentrations of Berg (25, 50, and 100 μM) at 24 hpi were detected by Western blotting (**G**) and IFA (**H**). Ribavirin was used as the positive control. E2 protein, green; nucleus, blue. Scale bar, 100 µm. The data show the mean ± SD for three independent experiments (* *p* < 0.05, ** *p* < 0.01, *** *p* < 0.001).

**Figure 2 viruses-16-01287-f002:**
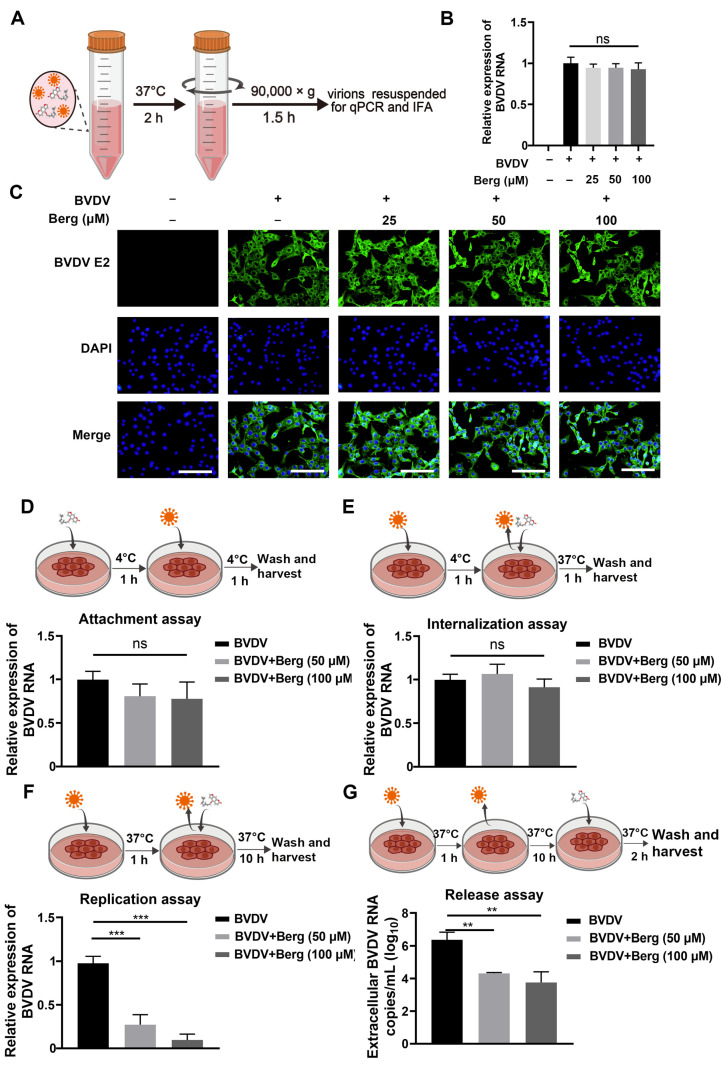
Berg inhibited BVDV infection by blocking the replication and release stage, rather than inactivating the virus particles. (**A**) Schematic representation of viral inactivation assay. BVDV was mixed with different concentrations of Berg and incubated for 1 h at 37 °C. The viruses were isolated after centrifugation and then infected MDBK cells for 24 h. Afterwards, total RNA was extracted for RT-qPCR and the cells were fixed for IFA to detect the virus infectivity. (**B**) The relative levels of BVDV RNA in the uninfected or BVDV-infected cells upon treatment with indicated concentrations of Berg were determined using RT-qPCR in the viral inactivation assay. (**C**) BVDV E2 protein levels were analyzed by IFA in the viral inactivation assay. E2 protein, green; nucleus, blue. Scale bar, 100 µm. (**D**–**G**) The effect of Berg on the (**D**) attachment, (**E**) internalization, (**F**) replication, and (**G**) release stage of the BVDV replication cycle. Viral RNA levels were analyzed by qRT-PCR. The data show the mean ± SD for three independent experiments (** *p* < 0.01; *** *p* < 0.001).

**Figure 3 viruses-16-01287-f003:**
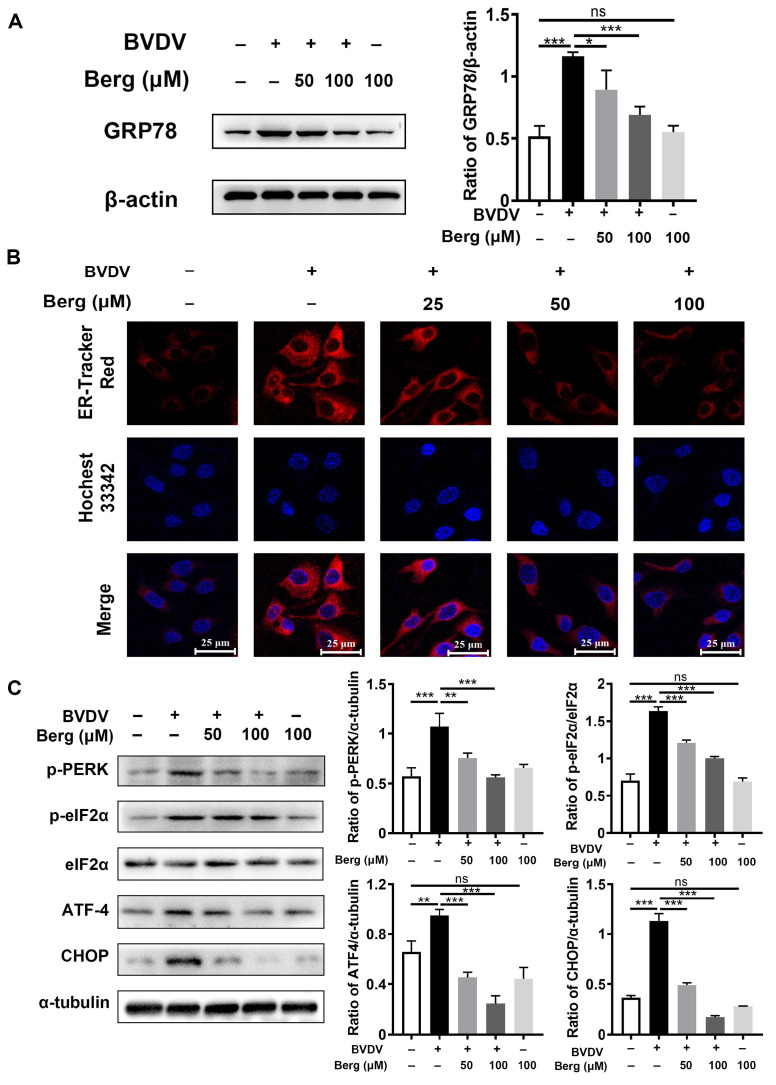
Berg attenuates BVDV-induced ER stress. The protein expressions of GRP78 (**A**), p-PERK, p-eIF2α, eIF2α, ATF-4, and CHOP (**C**) in the uninfected or BVDV-infected cells upon treatment with or without indicated concentrations of Berg (50 or 100 μM) were detected by Western blotting at 24 hpi. (**B**) Immunofluorescence staining of the ER tracker was conducted in the uninfected or BVDV-infected cells in response to various concentrations of Berg (50 or 100 μM). ER tracker, red; nuclei, blue. Scale bar, 25 µm. The data show the mean ± SD for three independent experiments (* *p* < 0.05; ** *p* < 0.01; *** *p* < 0.001).

**Figure 4 viruses-16-01287-f004:**
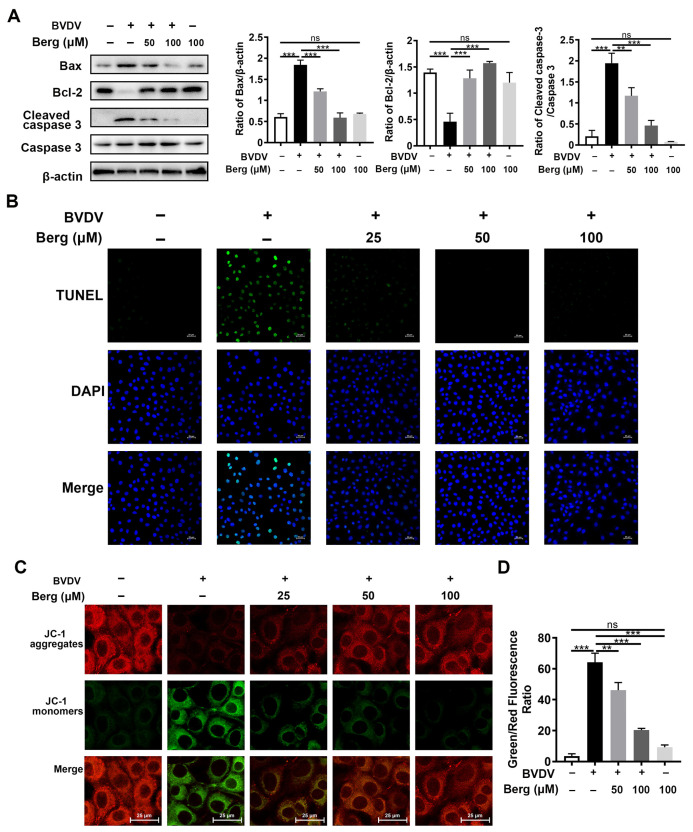
Berg inhibits BVDV-induced apoptosis. (**A**) Bax, Bcl-2, cleaved caspase 3, caspase 3 expressions were measured by Western blotting in the uninfected or BVDV-infected cells in response to different concentrations of Berg at 24 hpi. (**B**) TUNEL assay was conducted to assess the effects of Berg on BVDV-induced apoptosis in MDBK cells at 24 hpi. TUNEL: green. Nuclei: blue. Scale bar: 25 µm. (**C**) ∆*Ψm* was measured by JC-1 fluorescence staining in the uninfected or BVDV-infected cells in response to different concentrations of Berg at 24 hpi. (**D**) The results of JC-1 fluorescence staining are illustrated by the ratio of green/red fluorescence. The data show the mean ± SD for three independent experiments (** *p* < 0.01; *** *p* < 0.001).

**Figure 5 viruses-16-01287-f005:**
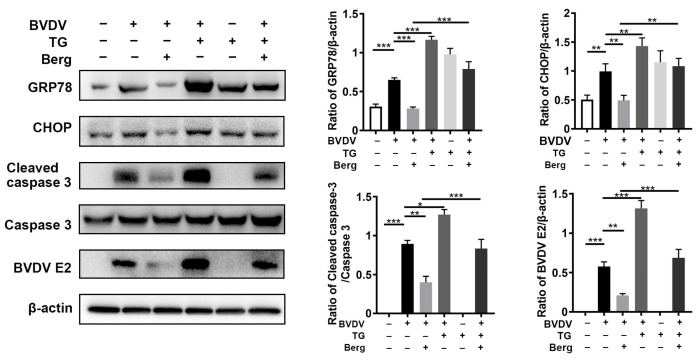
Berg suppresses the ER stress-mediated apoptosis in BVDV-infected cells. MDBK cells were pretreated with TG (200 μM) for 2 h, followed by BVDV infection in the presence or absence of Berg (100 μM). The expression levels of GRP78, CHOP, and cleaved caspase 3 were determined by Western blotting at 24 hpi. The graph presents the quantification analysis of GRP78, CHOP, cleaved caspase 3 and BVDV E2 levels. The data show the mean ± SD for three independent experiments (* *p* < 0.05; ** *p* < 0.01; *** *p* < 0.001).

**Figure 6 viruses-16-01287-f006:**
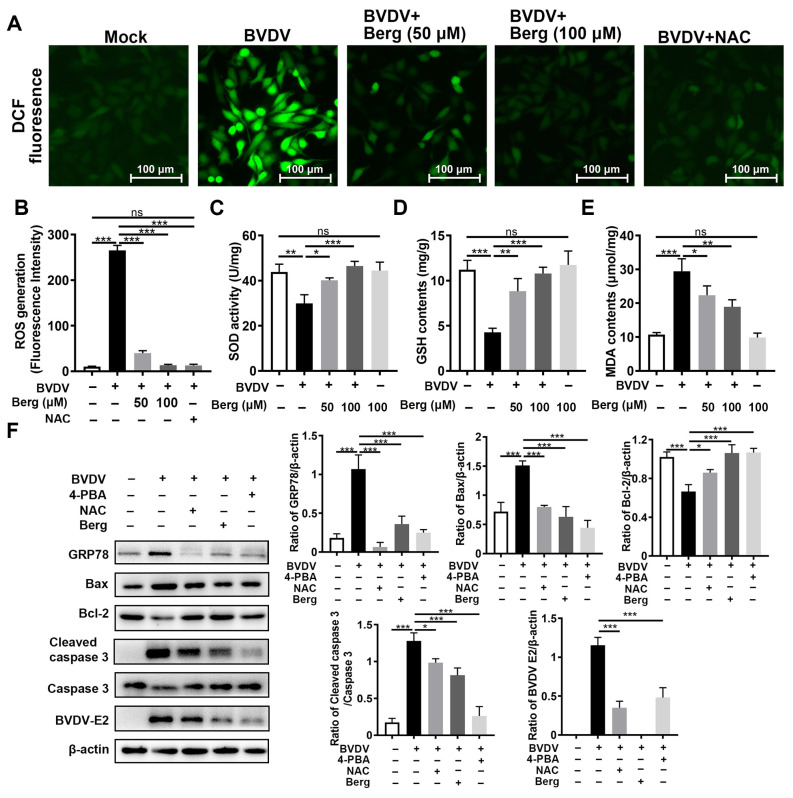
Berg relieves ER stress-mediated apoptosis by reducing ROS in BVDV-infected cells. (**A**) ROS levels were detected using a DCFH-DA kit. Scale bar: 100 µm. (**B**) Quantitative analysis of intracellular DCF fluorescence. The intracellular SOD (**C**), GSH (**D**), and MDA (**E**) levels were detected using kits. (**F**) The expression of BVDV E2, GRP78, Bax, Bcl-2, and cleaved caspase 3 proteins were measured by Western blotting in uninfected or BVDV-infected cells in response to different concentrations of 100 μM Berg, 100 μM NAC, or 2 mM 4-PBA at 24 hpi. The graph presents the quantification analysis of GRP78, Bax, Bcl-2, and cleaved caspase 3 levels. The data show the mean ± SD for three independent experiments (* *p* < 0.05; ** *p* < 0.01; *** *p* < 0.001).

**Figure 7 viruses-16-01287-f007:**
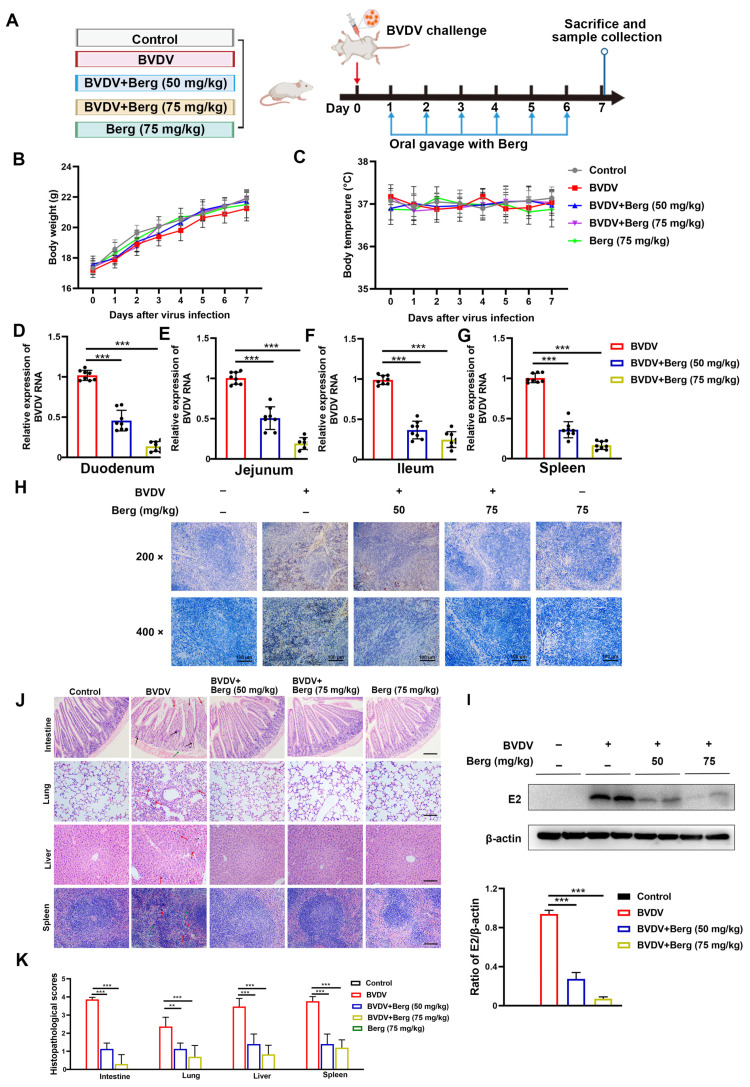
Berg inhibits virus replication and ameliorates pathological damages in BVDV-challenged mice. (**A**) Schematic diagram of the animal experiment. Body weight (**B**) and temperature (**C**) measurements of mice in each group (n = 8). The viral RNA levels of the duodenum (**D**), jejunum (**E**), ileum (**F**), and spleen (**G**) in BVDV-challenged mice were detected by RT-qPCR. The BVDV E2 protein expression of spleen in mice was determined by Western blotting (**H**) and IHC (**I**) assays. Scale bar: 100 µm. (**J**) Representative histopathological changes in intestine, lungs, liver, and spleen tissues in mice. Scale bar: 100 µm. (**K**) Scores of pathological changes in mice. The data show the mean ± SD for three independent experiments (** *p* < 0.01; *** *p* < 0.001).

**Figure 8 viruses-16-01287-f008:**
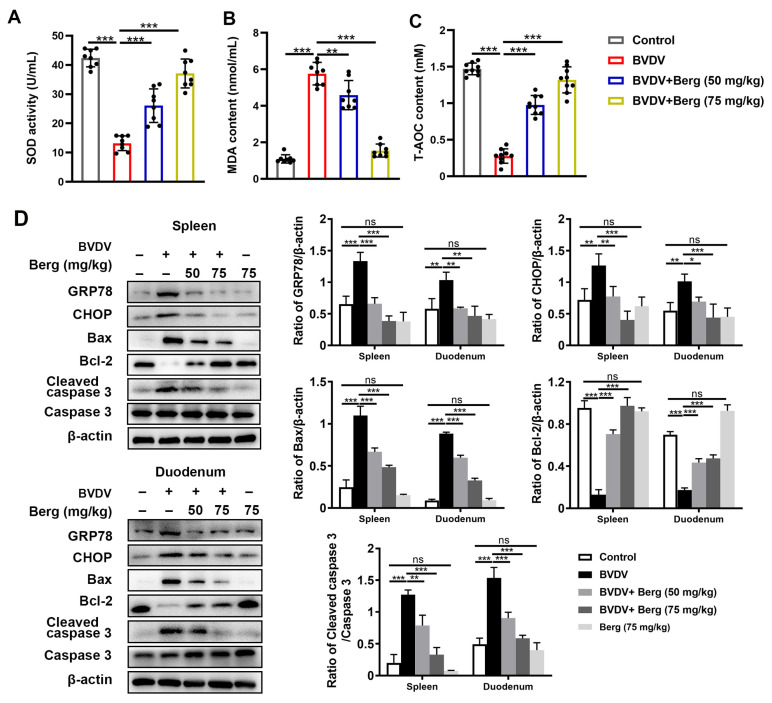
Berg weakens ER stress-mediated apoptosis through antioxidant system in BVDV-challenged mice The levels of SOD (**A**), MDA (**B**), and T-AOC (**C**) in the sera of mice were measured using commercial kits. (**D**) Western blotting analysis of GRP78, CHOP, Bax, Bcl-2, and cleaved caspase 3 expressions in the spleen and duodenum in mice. β-actin served as the loading control. The data show the mean ± SD for three independent experiments (* *p* < 0.05; ** *p* < 0.01; *** *p* < 0.001).

**Figure 9 viruses-16-01287-f009:**
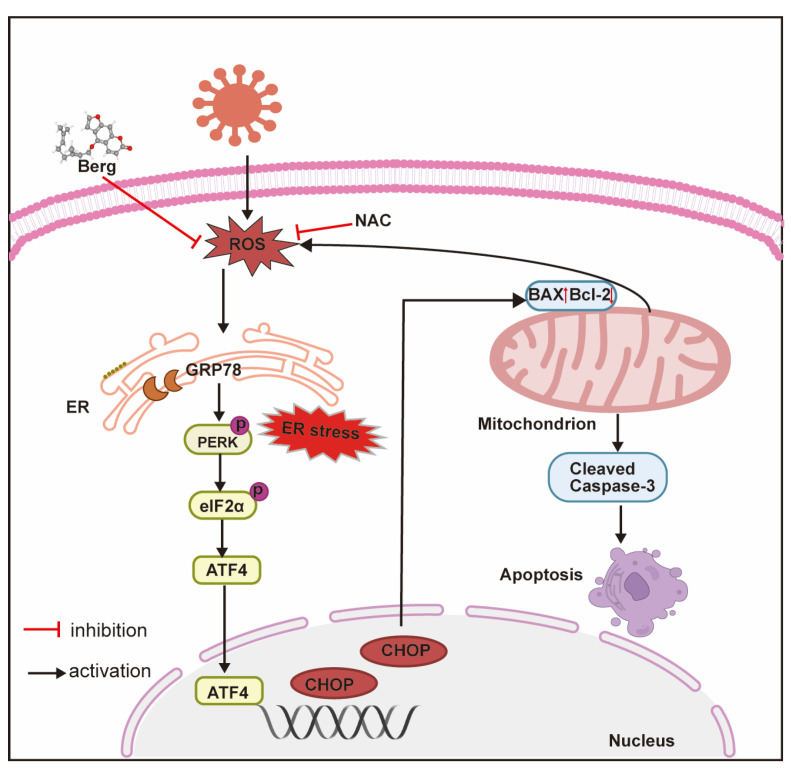
The main mechanism diagram of BVDV inhibition by Berg. Berg inhibits BVDV replication by suppressing the ER stress-mediated apoptosis via reducing ROS generation.

## Data Availability

Data are contained within the article.

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
