# Peer review of "Bergamottin Inhibits Bovine Viral Diarrhea Virus Replication by Suppressing ROS-Mediated Endoplasmic Reticulum Stress and Apoptosis"

_viruses, 2024, doi:10.3390/v16081287_

Round 1
Reviewer 1 Report (Previous Reviewer 1)
Comments and Suggestions for Authors
Accept in present form.
Reviewer 2 Report (Previous Reviewer 3)
Comments and Suggestions for Authors
Dear Authors, thank you for reviewing the manuscript and responding to the reviewers' comments.
The latest manuscript has been sufficiently improved.
This manuscript is a resubmission of an earlier submission. The following is a list of the peer review reports and author responses from that submission.
Round 1
Reviewer 1 Report
Comments and Suggestions for Authors
This study investigated the antiviral effect of Berg on BVDV infection using in vivo and in vitro models, and found that Berg inhibited BVDV replication through suppressing ER stress-mediated apoptosis via reduction of ROS generation. This study is meaningful and experimental design is appropriate. Additionally, the results support the conclusions. But the following suggestions or questions need to be addressed:
1. Line 19, “ER” and “ROS” should be fully spelled.
2. Line 31, “Pestivirus” and “Flaviviridae” should be italics.
3. What’s the source of BVDV 1-NADL strain?
4. The color in Figure 1B is not easily distinguishable. The figure with easily distinguishable patterns or colours for each group might make the results obvious.
5. Line 347, please note the concentration of TG in the legend of Figure 5.
6. Line 378, change “50 mg/kg or 75 mg/kg” to “50 or 75 mg/kg”, make this sentence concise.
7. Line 360, 4-PBA should be written in full when it first appears in the article.
8. Line 512, in the graphical abstract “the compound Berg” should be outside the cell.
Author Response
|
Thank you very much for taking the time to review this manuscript. Please find the detailed responses below and the corresponding revisions in the re-submitted files.
|
|
2. Point-by-point response to Comments and Suggestions for Authors |
|
Comments 1: Line 19, “ER” and “ROS” should be fully spelled.
|
|
Response 1: Thank you for pointing this out. We have made the change. The new sentence reads as follows: Mechanistically, Berg inhibits BVDV replication through suppressing endoplasmic reticulum (ER) stress-mediated apoptosis via reduction of reactive oxygen species (ROS) generation (page 1, line 19-20). |
|
Comments 2: Line 31, “Pestivirus” and “Flaviviridae” should be italics. |
|
Response 2: Thanks for your comments. The modification has been completed as suggested (page 1, line 33). |
|
Comments 3: What’s the source of BVDV 1-NADL strain?
|
|
Response 3: Thanks for your comments. The BVDV 1-NADL strain (GenBank Accession no. M31182.1) was purchased from the China Veterinary Culture Collection Center (Beijing, China) and stored in our laboratories. |
|
Comments 4: The color in Figure 1B is not easily distinguishable. The figure with easily distinguishable patterns or colours for each group might make the results obvious.
|
|
Response 4: Thanks for your comments. We agree with this comment. Therefore, we have modified the figure as you suggested (Figure 1B). |
|
Comments 5: Line 347, please note the concentration of TG in the legend of Figure 5.
|
|
Response 5: Thank you for pointing this out. The modification of the legend of Figure 5 has been completed as suggested. “MDBK cells were pretreated with TG (200 μM) for 2 h, followed by infection with BVDV in the presence or absence of Berg”. (page 11, line 377) |
|
Comments 6: Line 378, change “50 mg/kg or 75 mg/kg” to “50 or 75 mg/kg”, make this sentence concise.
|
|
Response 6: Thanks for your comments. The modification has been completed as suggested (page 12, line 412). |
|
Comments 7: Line 360, 4-PBA should be written in full when it first appears in the article.
|
Response 7: Thank you for pointing this out. The “4-PBA” has been changed into “4-phenylbutyric acid (4-PBA)” (page 11, line 392). |
|
Comments 8: Line 512, in the graphical abstract “the compound Berg” should be outside the cell.
|
|
Response 8: Thanks for your comments. Changes were made as suggested. |

Reviewer 2 Report
Comments and Suggestions for Authors
General Comments: The effects of Bergamottin (Berg), a natural furanocoumarin, on BVDV replication in MDBK cells is examined in this manuscript. However, the manuscript needs a more detailed and clearly written description of the experimental designs and methods and extensive editing for better reader comprehension.
Specific Comments:
Introduction:
1) lines 59-60, paragraph 59-67 – This paragraph needs to be edited in for clarity.
2) Check for typos/misspelling, e.g., Intro line 71 “re-ports” should be reports and Line 72 “dis-play” should be display.
Methods:
1. 2.1 – Were the MDBKs and fetal bovine serum tested for adventitial ncp BVDV?
2. 2.2 – were the controls with dilutions of media containing DMSO included?
3. 2.2 – How was cell viability calculated from the absorbance readings?
4. 2,.2 – The experimental design is unclear. Why was the antiviral drug ribavirin used with Berg in this experiment? How would the authors distinguish between the antiviral activity of ribavirin and that of Berg?
5. 2.3 – Is there a reference for the BVDV primers used? If not, how were the primers designed? What was the primer efficiency in MDBKs?
6. 2.3 - Show the calculation of the comparative Ct method. The comparative Ct method should be referenced.
7. 2.3 – What conditions (i.e., temperatures, cycles, etc) were used for the RTqPCR?
8. 2.4 – line 124 – What was the source of the rabbit anti-E2 antibody?
9. 2.4 – What is the make/model of the confocal microscope? What software was used for imaging?
10. 2.5 – the source/company-location of the primary and secondary antibodies, the fluor and the imaging software should be listed.
11. 2.6 – line 145. What concentrations of Berg (or a range) were tested?
12. 2.6 – This section is very unclear and the experimental design difficult to follow, e.g., line 148 – What exactly were the samples collected? How would one quantify “viral particles” using these methods?
13. 2.6 – Lines 150-151. How were these concentrations of Berg selected?
14. 2.6 – Lines. MDBKs are adherent cells. How were they “collected” without affecting virions attached to the cell surface? How does RTqPCR data correspond to viral attachment? This doesn’t make sense as written.
15. 2.6 – How is viral “mRNA” distinguished from viral RNA in the RTqPCR method since BVDV has a positive sense RNA genome with 5’ and 3’ UTR and does not have a poly-A tail?
16. 2.6 – line 157. How was the removal of attached virions confirmed?
17. 2.6 – line 163 – What mRNAs were being measured?
18. 2.9 – What constituted mock-infection?
19. 2.10 – What product/companies/location were used? Lacks sufficient detail.
20. 2.12 – Experimental design seems to be missing at least one control group.
21. 2.12 – Did the control mice receive PBS with the same concentration of DMSO as the Berg-treated mice?
22. 2.13 – This section lacks important details such as the deparaffinization, antigen-retrieval, enzyme used, products/companies/locations, incubation times and temperatures. Staining mouse tissue sections with mouse monoclonal antibodies are fraught with problems. None are addressed here.
Results:
1. 3.1 – lines 232-233. The concentrations stated here do not match those in the methods.
2. 3.1 – line 236. BVDV isn’t exactly mRNA – basic genome information.
3. Figure 2 Legend: What does “disturb” really mean? Do you mean decrease viral replication as in the number of progeny viruses produced per cell?
a. A, B, C, D, E) Neither the figures or the text really describe the technique sufficiently.
References: There are multiple publications regarding testing antivirals in BVDV and other members of the Flavivirus family. At least a few relevant references should be read by the authors and included here.

/
Author Response
|
Response to Reviewer 2 Comments
|
||
|
1. Summary |
|
|
|
Thank you very much for taking the time to review this manuscript. Please find the detailed responses below and the corresponding revisions in the re-submitted files.
|
||
|
2. Point-by-point response to Comments and Suggestions for Authors |
||
|
Introduction: Comments 1: lines 59-60, paragraph 59-67 – This paragraph needs to be edited in for clarity.
|
||
|
Response 1: Thanks for your comments. We have re-written the paragraph to be more concise as suggested (page 2, line 65-75). Endoplasmic reticulum (ER) is a crucial cellular membrane organelle in eukaryotic cells. Viruses naturally employ the host's translation machinery to produce an extensive amount of viral proteins in the ER lumen, which will disturb the ER homeostasis and ul-timately results in unavoidable ER stress [21]. To cope with the deleterious effects of ER stress, the unfolded protein response (UPR) is triggered to restore ER homeostasis [22]. Mounting evidence suggests that ER stress and persistent UPR play a pivotal role in the pathogenesis of viral infection disease, especially ER-tropic viruses such as Flaviviridae family [23-25]. In the case of BVDV infection, the cellular ER stress and UPR pathway are activated to maintain a favorable environment for its propagation [26-27]. Previous research has demonstrated that Berg could mitigate ER stress in cafeteria diet-fed mice [28]. Nevertheless, it remains unclear whether Berg affects ER stress induced by BVDV. |
||
|
Comments 2: Check for typos/misspelling, e.g., Intro line 71 “re-ports” should be reports and Line 72 “dis-play” should be display. |
||
|
Response 2: We were really sorry for our careless mistakes. Thanks for your reminder. We have corrected the words “re-ports” and “dis-play” into “reports” and “display” (page 2, line 79, 80).
Methods: |
||
|
Comments 1: 2.1 – Were the MDBKs and fetal bovine serum tested for adventitial ncp BVDV?
|
||
|
Response 1: Thank you for pointing this out. Yes, we did. Because BVDV is a major source of contamination of fetal bovine serum and cell lines, so we usually detect the presence of BVDV at the beginning of cell culture to ensure the accuracy of the experiments. |
||
|
Comments 2: 2.2 – were the controls with dilutions of media containing DMSO included?
|
||
|
Response 2: Thanks for your comments. For the cell viability experiment, cells treated with 0.1% DMSO were used as the negative control. In vitro drug treatments, Berg was dissolved in DMSO (below 0.1%). |
||
|
Comments 3: 2.2 – How was cell viability calculated from the absorbance readings?
|
||
|
Response 3: Thank you for pointing this out. The calculated formula of the cell viability was added in page 3, line 111. The cell viability was calculated as following formula: Cell viability (%) = [A (experimental group) – A (blank group) / A (control group) – A (blank group)] × 100%, A (experimental group) contains the cells, CCK-8 reagent and the drug, the blank group contains medium and CCK-8 reagent but does not contain cells and drug, and the control group contains cells and CCK-8 reagent but does not contain the tested drug. |
||
|
Comments 4: 2.2 – The experimental design is unclear. Why was the antiviral drug ribavirin used with Berg in this experiment? How would the authors distinguish between the antiviral activity of ribavirin and that of Berg?
|
||
|
Response 4: Thanks for your comments. Because ribavirin was a well-known inhibitor of viral RNA polymerase, which was reported to have broad-spectrum antiviral activities against RNA virus infections [1,2]. Thus, we used ribavirin as a positive antiviral drug control. As shown in Figure 1, 100 µM Berg reduced the viral BVDV RNA and protein levels more significantly than ribavirin, suggesting that the antiviral effect of Berg can be comparable to that of ribavirin. [1] Markland, W., McQuaid, T.J., Jain, J., Kwong, A.D., 2000. Broad-spectrum antiviral activity of the IMP dehydrogenase inhibitor VX-409: a comparison with ribavirin and demonstration of antiviral additivity with alpha interferon. Antimicrob. Agents Chemother. 44, 859–866. |
||
|
[2] Ferenci P, Bernstein D, Lalezari J, Cohen D, Luo Y, Cooper C, Tam E, Marinho RT, Tsai N, Nyberg A, Box TD, Younes Z, Enayati P, Green S, Baruch Y, Bhandari BR, Caruntu FA, Sepe T, Chulanov V, Janczewska E, Rizzardini G, Gervain J, Planas R, Moreno C, Hassanein T, Xie W, King M, Podsadecki T, Reddy KR. 2014. ABT-450/r-ombitasvir and dasabuvir with or without ribavirin for HCV. N Engl J Med 370:1983–1992.
Comments 5: 2.3 – Is there a reference for the BVDV primers used? If not, how were the primers designed? What was the primer efficiency in MDBKs?
Response 5: Thanks for your comments. The BVDV 5’UTR primers were set according to the citation 27. The primers were specific and the product is a fragment of 155 bp. We often use them to identify BVDV[27]. [27]Wang, J.; Chen, K. Y.; Wang, S. H.; Liu, Y.; Zhao, Y. Q.; Yang, L.; Yang, G. H.; Wang, X. J.; Zhu, Y. H.; Yin, J. H.; Wang, J. F. Effects of Spatial Expression of Activating Transcription Factor 4 on the Pathogenicity of Two Phenotypes of Bovine Viral Diarrhea Virus by Regulating the Endoplasmic Reticulum-Mediated Autophagy Process. Microbiol Spectr 2023, 11, e0422522.
Comments 6: 2.3 - Show the calculation of the comparative Ct method. The comparative Ct method should be referenced.
|
||
|
Response 6: Thanks for your comments. The calculation of the comparative Ct method was showed as follows: GAPDH was used the reference gene. The threshold cycle (ΔCt) value was calculated as ΔCt = Ct(5’UTR)-Ct(GAPDH). The relative gene expression levels were calculated by the 2-ΔΔCt method (page 3, line 127-128). |
||
|
Comments 7: 2.3 – What conditions (i.e., temperatures, cycles, etc) were used for the RT-qPCR?
|
|
Response 7: Thanks for your comments. The conditions for RT-qPCR have been added in page 3, line 125-126. The RT-qPCR was performed using the following procedure: 95°C for 30 s, and then 45 cycles of 95°C for 5 s and 60°C for 30 s. |
|
Comments 8: 2.4 – line 124 – What was the source of the rabbit anti-E2 antibody?
|
|
Response 8: Thank you for pointing this out. The rabbit anti-E2 antibody was prepared in our laboratory and the related information has been added in page 4, line 145. |
|
Comments 9: 2.4 – What is the make/model of the confocal microscope? What software was used for imaging?
|
|
Response 9: Thank you for pointing this out. Images of the staining samples were acquired using LASX software and a confocal laser scanning microscope (Leica SP8, Wetzlar, Germany) (page 4, line 150). |
|
Comments 10: 2.5 – the source/company-location of the primary and secondary antibodies, the fluor and the imaging software should be listed.
|
|
Response 10: Thanks for your comments. The information of the source/company-location of the primary and secondary antibodies, the fluor and the imaging software has been listed as requested (page 4, line 159-166). GRP78 (1:1000) from Wanlei Bio (Shenyang, China), phosphor-PERK (1:1000), phospho-eIF2α (1:1000), eIF2α (1:5000), ATF-4 (1:1000), CHOP (1:1000), Bax (1:1000), Bcl-2 (1:1000), Caspase 3 (1:1000), Cleaved caspase 3 (1:1000), α-tubulin (1:20000) and β-actin (1:20000) from Cell Signaling Technology (Danvers, USA). After washing, the membranes were incubated with anti-rabbit IgG or anti-mouse IgG from Proteintech Group, Inc. (Rosemont, USA) for 1 h at 37°C. Images of target protein bands were captured using a Tanon 6200 chemiluminescence imaging workstation and analyzed using Image J software. |
|
Comments 11: 2.6 – line 145. What concentrations of Berg (or a range) were tested?
|
|
Response 11: Thanks for your comments. The concentrations of Berg in the virus inactivation assay were 25, 50, or 100 μM (page 4, line 169). |
|
Comments 12: 2.6 – This section is very unclear and the experimental design difficult to follow, e.g., line 148 – What exactly were the samples collected? How would one quantify “viral particles” using these methods?
|
|
Response 12: Thank you for pointing this out. We have revised the text to address your concern and hope that it is now clearer (page 4, line 172-173). The isolated viruses were then suspended in cell culture medium and infect MDBK cells for 12 h. Afterwards, total RNA were extracted for RT-qPCR and the cells were fixed for IFA assay to detect the virus infectivity. |
|
Comments 13: 2.6 – Lines 150-151. How were these concentrations of Berg selected?
|
|
Response 13: Thanks for your comments. In the antiviral assays, 50 and 100 μM Berg significantly inhibited BVDV replication without measurable cytotoxicity. Thus, we select these concentrations of Berg. |
|
Comments 14: 2.6 – Lines. MDBKs are adherent cells. How were they “collected” without affecting virions attached to the cell surface? How does RTqPCR data correspond to viral attachment? This doesn’t make sense as written.
|
|
Response 14: Thanks for your comments. In the viral attachment assay, cells were pretreated for with Berg (50 or 100 μM) for 1 h, after which BVDV was added in cold medium in the presence or absence of Berg. Incubation was continued for 2 h at 4°C to maximize virus-cell binding. At this temperature, virions can attach to their receptors on the cell surface, but virus internalization is hampered due to the inhibition of intracellular transport [1,2]. Therefore, the RT-qPCR data corresponds to the viral attachment. In the viral attachment assay, unbound virions were removed after washing with ice-cold PBS while those binding the cells were not washed away. Then we used trizol to extract the total RNA for RT-qPCR assay to detect the viral RNA levels. |
|
[1] Wang G, Hernandez R, Weninger K, Brown DT. 2007. Infection of cells by Sindbis virus at low temperature. Virology 362:461– 467. [2] Harden EA, Falshaw R, Carnachan SM, Kern ER, Prichard MN. 2009. Virucidal activity of polysaccharide extracts from four algal species against herpes simplex virus. Antiviral Res 83:282–289.
Comments 15: 2.6 – How is viral “mRNA” distinguished from viral RNA in the RTqPCR method since BVDV has a positive sense RNA genome with 5’ and 3’ UTR and does not have a poly-A tail?
|
|
Response 15: Thanks for your comments. The 5'UTR is a highly conserved sequence, and is the most commonly used region for identifying BVDV. In the RT-qPCR experiment, the extracted total RNA and the specific primers of 5’UTR were used. Thus, the relative levels of 5’UTR can indicate the viral RNA level. Additionally, we have replaced “mRNA level” with “viral RNA levels” in the manuscript to make it more precise. |
|
Comments 16: 2.6 – line 157. How was the removal of attached virions confirmed?
|
|
Response 16: Thanks for your comments. The cells were treated with citric acid buffer at pH 3 at room temperature, that can inactivate the virus that bound but had not penetrated the cell membrane. The experiments were conducted according to the following studies [1,2]. [1] Wang P, Bai J, Liu X, Wang M, Wang X, Jiang P. Tomatidine inhibits porcine epidemic diarrhea virus replication by targeting 3CL protease. Vet Res. 2020 Nov 11;51(1):136. [2] Suo X, Wang J, Wang D, Fan G, Zhu M, Fan B, Yang X, Li B. DHA and EPA inhibit porcine coronavirus replication by alleviating ER stress. J Virol. 2023 Nov 30;97(11):e0120923. |
|
Comments 17: 2.6 – line 163 – What mRNAs were being measured?
|
|
Response 17: Thanks for your comments. For the virus release assay, the supernatants were harvested to analyze viral RNA levels. |
|
Comments 18: 2.9 – What constituted mock-infection?
|
|
Response 18: Thanks a lot for your comments. |
|
We apologize for the confusion and revise the sentence. MDBK cells were infected with BVDV (MOI = 1) and treated with various concentrations of Berg (25, 50, or 100 μM) for 24 h (page 5, line 206-207).
Comments 19: 2.10 – What product/companies/location were used? Lacks sufficient detail.
|
|
Response 19: Thanks a lot for your comments. We have added the information of product/companies/location as requested (page 5, line 214-216). |
|
Comments 20: 2.12 – Experimental design seems to be missing at least one control group.
|
|
Response 20: Thanks a lot for your comments. We have revised and labelled the information of control group (page 5, line 235-237). |
|
Comments 21: 2.12 – Did the control mice receive PBS with the same concentration of DMSO as the Berg-treated mice?
|
|
Response 21: Thanks a lot for your comments. Yes. Berg was dissolved with 0.5% DMSO in vivo experiments, so mice in the control group were treated with an equal volume of PBS with 0.5% DMSO. |
|
Comments 22: 2.13 – This section lacks important details such as the deparaffinization, antigen-retrieval, enzyme used, products/companies/locations, incubation times and temperatures. Staining mouse tissue sections with mouse monoclonal antibodies are fraught with problems. None are addressed here.
|
|
Response 22: Thanks a lot for your comments. The details were added in 2.13 as suggested. |
|
The viral antigens in the spleen were examined by IHC. Briefly, spleen sections were dewaxed and antigens were retrieved with citrate buffer (pH 6.0). Afterward, the endogenous peroxidase was blocked by 3% hydrogen peroxide for 10 min at room temperature. Subsequently, the sections were blocked with 10% goat serum for 30 min at room temperature and incubated with rabbit anti-BVDV E2 polyclonal primary antibody (prepared in our laboratory) overnight at 4°C. Then the HRP-conjugated goat anti-rabbit IgG was added to the slides and incubated for 20 min at 37°C. Following color development using 3,3’-diaminobenzidine (DAB, Zhongshan Golden Bridge Biotechnology, Beijing, China), images of sections were viewed under a light microscope. Additionally, we have revised the information of the primary antibody and corrected it (page 5, line 245-251).
Results: Comments 1: 3.1 – lines 232-233. The concentrations stated here do not match those in the methods.
|
|
Response 1: Thanks a lot for your comments. We have modified the methods accordingly (page 3, line 115). |
|
Comments 2: 3.1 – line 236. BVDV isn’t exactly mRNA – basic genome information.
|
|
Response 2: Thanks a lot for your comments. We have revised the description in the results 3.1, replacing “BVDV 5’UTR mRNA” with “the viral RNA levels” (page 7, line 291). |
|
Comments 3: Figure 2 Legend: What does “disturb” really mean? Do you mean decrease viral replication as in the number of progeny viruses produced per cell? A, B, C, D, E) Neither the figures or the text really describe the technique sufficiently.
|
|
Response 3: Thanks a lot for your comments. The word “disturb” in Figure 2 legend means “Berg inhibited BVDV infection by blocking the replication and release stage”. In order to make the description precise, we have modified it as your suggestion. Additionally, the sufficient details of Figure 2A-E were added (page 8, line 305-309). |
References: There are multiple publications regarding testing antivirals in BVDV and other members of the Flavivirus family. At least a few relevant references should be read by the authors and included here.
Response 1: Thanks a lot for your comments. According to your suggestion, a few relevant references have been added in the introduction (page 2, line 47-52, citations 11-14).

Reviewer 3 Report
Comments and Suggestions for Authors
Author Response
|
Response to Reviewer 3 Comments
|
||||
|
1. Summary |
|
|
||
|
Thank you very much for taking the time to review this manuscript. Please find the detailed responses below and the corresponding revisions in the re-submitted files.
|
||||
|
2. Point-by-point response to Comments and Suggestions for Authors |
||||
|
Comments 1: I suggest you add fully experimental details and present completely all the results.
|
||||
|
Response 1: Thanks a lot for your comments. We agree with this comment and make some modifications as suggested. Therefore, we have added the sufficient experimental details in the part of materials and methods and also presented a clearer description of the results. For example, the calculation method of cell viability, the detail of comparative CT method, the reaction procedure of RT-qPCR (i.e., temperatures, cycles, etc), the make/model of the confocal microscope, the source/company-location of the primary and secondary antibodies, the fluor and the imaging software and so on.
Introduction |
||||
|
Comments 2: This section and elsewhere as needed, the manuscript needs to be edited to reflect the current taxonomy adopted by ICTV whereby Pestivirus A, B, C and through to K have been replaced by binomial species names eg Pestivirus bovis, Pestivirus Taurus, pestivirus suis etc.
|
||||
|
Response 2: Thanks a lot for you comments. We agree with this comment. Therefore, we modified the species with the current taxonomy in the manuscript and marked red (page 1, line 30).
|
||||
|
Comments 3: Which genotypes are most prevalent in the area?
|
||||
|
Response 3: Thanks a lot for your comments. Currently, BVDV-1a, 1b, and 1m are the major prevalent genotypes in China, and BVDV-1v, an emerging genotype in China, is also a regional prevalence [1-4]. [1] Deng M, Ji S, Fei W, Raza S, He C, Chen Y, et al. Prevalence study and genetic typing of bovine viral diarrhea virus (Bvdv) in four bovine species in China. PLoS ONE. (2015) 10: e0121718. [2] Deng M, Chen N, Guidarini C, Xu Z, Zhang J, Cai L, et al. Prevalence and genetic diversity of bovine viral diarrhea virus in dairy herds of China. Vet Microbiol. (2020) 242:108565. [3] Tian B, Cai D, Li W, Bu Q, Wang M, Ye G, et al. Identification and genotyping of a new subtype of bovine viral diarrhea virus 1 isolated from cattle with diarrhea. Arch Virol. (2021) 166:1259–62. [4] Shi H, Li H, Zhang Y, Yang L, Hu Y, Wang Z, et al. Genetic diversity of bovine pestiviruses detected in backyard cattle farms between 2014 and 2019 in Henan Province, China. Front Vet Sci. (2020) 7:197.
Materials and Methods |
||||
|
Comments 4: what is the title of the BVDV-1 NADL strain?
|
||||
|
Response 4: Thanks a lot for your comments. The BVDV 1-NADL strain (GenBank Accession Number M31182.1) was a standard and cytopathic strain and purchased from the China Veterinary Culture Collection Center (Beijing, China).
|
||||
|
Comments 5: why was only one strain used and not several?
|
||||
|
Response 5: Thanks a lot for your comments. Because BVDV-1 is the major prevalent genotype in China, so we choose the standard BVDV-1 strain to investigate the antiviral effect and the corresponding mechanism of Berg. Additionally, the related experiments about the antiviral effect and mechanism of Berg against noncytopathic strains are being carried out and have not yet been completed. In the further, we will conduct the experiments using multiple strains as you suggested.
|
||||
|
Comments 6: Why was BVDV-2 not considered?
|
||||
|
Response 6: Thanks a lot for your comments. According to the relevant studies, currently BVDV-1 is the major prevalent genotype in China, especially the BVDV-1a, 1b, 1c and 1m subgenotypes [1-3]. Therefore, BVDV-1 was chosen as our study subject, and in the future we will also conduct related studies on BVDV-2 if necessary. [1] Chang L, Qi Y, Liu D, Du Q, Zhao X, Tong D. Molecular detection and genotyping of bovine viral diarrhea virus in Western China. BMC Vet Res. 2021 Feb 2;17(1):66. [2] Wu Y, Zhang G, Jiang H, Xin T, Jia L, Zhang Y, Yang Y, Qin T, Xu C, Cao J, Ameni G, Ahmad A, Ding J, Li L, Ma Y, Fan X. Molecular Characteristics of Bovine Viral Diarrhea Virus Strains Isolated from Persistently Infected Cattle. Vet Sci. 2023,10(7):413. [3] Zhu J, Wang C, Zhang L, Zhu T, Li H, Wang Y, Xue K, Qi M, Peng Q, Chen Y, Hu C, Chen X, Chen J, Chen H, Guo A. Isolation of BVDV-1a, 1m, and 1v strains from diarrheal calf in china and identification of its genome sequence and cattle virulence. Front Vet Sci. 2022,16(9):1008107.
|
||||
|
||||
|
Comments 8: how many replicates included for each treatment?
|
||||
|
Response 8: Thanks a lot for your comments. Three replicates included for each treatment in our study and the experimental data were obtained from three independent experiments for quantitative analyses.
|
||||

Reviewer 4 Report
Comments and Suggestions for Authors
In their study the authors investigate the effect of Bergamottin on the replication of the cytopathogenic BVDV-1 strain NADL in bovine MDBK cells and in a mouse model. The authors demonstrate that Bergamottin (Berg) reduces viral replication on the level of the viral genomic RNA by approximately 95% (resolution in Fig. 1 does not allow a more precise assessment) and thus at a level similar to the one of Ribavirin. They experimentally exclude a direct inactivation of the BVD virions by Berg as well as a disturbance of the binding step. In the following experiments the authors demonstrate that the dose-dependent inhibition of BVDV correlates with the inhibition of the BVDV-induced effects on the MDBK cells, which are BVDV-induced ER-stress (p-PERK, p-IEF2alpha, ATF4, CHOP) as well as BVDV-induced apoptosis by ROS-dependent ER stress. Finally, the authors inoculated mice intraperitoneal with BVDV; mice were treated orally with Berg. This treatment reduced viral E2 protein expression and viral RNA titres in the mouse significantly. Along these lines, the tissue damage observed in BVDV infected mice was ameliorated upon Berg treatment. GRP78, CHOP, Bax and cleaved Caspase 3 were elevated in BVDV inoculated mice, which was reversed upon Berg treatment. According to the authors, the latter finding indicates that Berg weakens ER stress-mediated apoptosis induction.
The general finding (Fig. 1) that RNA replication of BVDV-1 strain NADL is strongly inhibited by Berg in MDBK cells and in mice (Fig. 7) is of interest. The RT-PCR data are clear but the titres of infectious BVDV particles from MDBK cells with and without Berg addition would be a valuable and highly relevant addition since only these data will allow the reader to judge how massive the inhibition potential of Berg is. A reduction by a factor of 10 does not indicate elimination of virus replication for a virus with e.g. 106 infectious particles per ml.
In the following experiments the authors show that a cytopathogenic BVDV strain NADL induces ER stress and apoptosis. These facts have been reported before and thus are not novel. The reversal of the effects of a viral infection on cells by an antiviral that inhibits viral replication is to be expected. A potent inhibitor of the viral RNA polymerase would also reverse the effects of viral infection on the cell. To clarify the mechanistic behind the inhibition by Berg the authors need to show whether Berg does inhibit these cellular pathways also in the absence of viral infection to sort cause and effect. For example: How much BVDV titre reduction is caused by the addition of NAC? In Fig. 6F the E2 levels stay constant, indicating that the reduction of ROS by NAC does not inhibit BVDV replication. This needs to be clarified experimentally.
Finally, in line 450 the authors conclude “These findings…laying the groundwork for potential clinical applications”. The authors used throughout their study a cytopathogenic variant of the pestivirus species BVDV-1. The cp virus mutants do induce apoptosis. However, the BVDV isolates relevant in the field are selectively noncp BVDV strains. Only noncp BVDV can establish persistent infections. To judge whether Berg also inhibits noncp BVDV, which does not induced apoptosis, and whether inhibition of noncp BVDV is related to ER-stress reduction has to be shown by the authors before they can draw conclusion on “potential clinical applications”.
While the described effect of Bergamottin on the replication on cp BVDV-1 strain NADL in MDBK cells and in the mouse model are clearly of interest for the reader of Viruses. However, the investigation of the mechanistic background of inhibition needs to be improved. At the moment, the conclusions drawn by the authors are not justified by the experiments.
Points of criticism:
1. In contrast to the statement of the authors (e.g. line 2 of the abstract) efficient vaccines are available against BVDV-1 and-2. Thus, the statement in this form is not valid.
2. Please do not use “BVDV 5´mRNA levels” (throughout text); this term is misleading. The RT-PCR method applied in this study determines the number of viral RNA genome copies (which serve in part as viral mRNA) but there is no 5´mRNA (or any subgenomic viral RNA) in those cells.
3. Fig. 2A: please indicate that the virus is used after centrifugation for an infection and RT qPCR is done 24 hpi.
4. The finding that RNA replication of BVDV is strongly inhibited by Berg is of interest. The reduction of the infectious titre of BVDV produced in MDBK cells by Berg needs to be determined to get a better understanding of the inhibitory potential of Berg.
5. Are the pathways induced by BVDV blocked by Berg also in the absence of infection? This needs to be shown to prove the author`s statement “Berg inhibits BVDV replication through suppressing ER stress-mediated apoptosis via reduction of ROS generation.” This is required to clarify cause and effect (also see above).
6. Line 438: Citation 31 does not deal with BVDV infection and is thus not suited.
7. Indicated in Fig. 6 F (table) the concentration of Berg, NAC and 4-PBA.
8. Inhibition of noncp BVDV and the underlying mechanism needs to be investigated when conclusions on a clinical potential of Berg shall be drawn.
9. The authors claim a potential inhibition at the “release step”. This reviewer sees no valid data supporting that statement. The inhibition of apoptosis by Berg (line 483) is no valid argument since noncp BVDV is released from cells also in the absence of apoptosis.
Minor points:
Line 59: “Flavivirus, the member of ER-tropic virus”; please change this sentence into better readable form. The cited paper deals with members of the genus “Flavivirus”; these viruses differ massively from members of the genus pestivirus. Thus, the citation seems not suited.
Line 439: “released stages” should read “release stage”.
Line 452: “Viruses interacted with” should read “Viruses interact with”.
Comments on the Quality of English LanguageMinor corrections required.
Author Response
|
Response to Reviewer 4 Comments
|
||
|
1. Summary |
|
|
|
Thank you very much for taking the time to review this manuscript. Please find the detailed responses below and the corresponding revisions in the re-submitted files.
|
||
|
2. Point-by-point response to Comments and Suggestions for Authors |
||
|
Comments 1: In contrast to the statement of the authors (e.g. line 2 of the abstract) efficient vaccines are available against BVDV-1 and-2. Thus, the statement in this form is not valid.
|
||
|
Response 1: Thanks a lot for your comments. We have revised the related descriptions in the abstract. The revised text reads as follows. Vaccines usually provide limited efficacy against BVDV due to the emergence of mutant strains. Therefore, developing novel strategies to combat BVDV infection is urgently needed (page1, line12-14).
|
||
|
Comments 2: Please do not use “BVDV 5´mRNA levels” (throughout text); this term is misleading. The RT-PCR method applied in this study determines the number of viral RNA genome copies (which serve in part as viral mRNA) but there is no 5´mRNA (or any subgenomic viral RNA) in those cells.
|
||
|
Response 2: Thanks a lot for your comments. We have corrected “BVDV 5’UTR mRNA level” to “viral mRNA level” throughout the text and the figures as suggested.
|
||
|
Comments 3: Fig. 2A: please indicate that the virus is used after centrifugation for an infection and RT qPCR is done 24 hpi.
|
||
|
Response 3: Thanks a lot for your comments. The modification has been completed as suggested. The viruses were isolated after centrifugation and then infected MDBK cells for 24 h. Afterwards, total RNA was extracted for RT-qPCR and the cells were fixed for IFA assay to detect the virus infectivity (page 8, line 306-310).
|
||
|
Comments 4: The finding that RNA replication of BVDV is strongly inhibited by Berg is of interest. The reduction of the infectious titre of BVDV produced in MDBK cells by Berg needs to be determined to get a better understanding of the inhibitory potential of Berg.
|
|
Response 4: Thanks a lot for your comments. The infectious titre of BVDV produced in MDBK cells has been added to further characterize the antiviral effect of Berg, please see Figure 1 F (page 7).
|
|
Comments 5: Are the pathways induced by BVDV blocked by Berg also in the absence of infection? This needs to be shown to prove the author`s statement “Berg inhibits BVDV replication through suppressing ER stress-mediated apoptosis via reduction of ROS generation.” This is required to clarify cause and effect (also see above).
|
|
Response 5: Thanks a lot for your comments. In order to investigate the effect of Berg on the signaling pathways induced by BVDV, five groups were set up: mock group, BVDV group, BVDV + Berg (50 μM) group,BVDV + Berg (100 μM) group and Berg (100 μM) group. It was found that the individual administration of Berg affected the activation of the related pathways, but the effect was not significant compared to the control group.
|
|
Comments 6: Line 438: Citation 31 does not deal with BVDV infection and is thus not suited.
|
|
Response 6: Thanks a lot for your comments. We have checked the literature carefully and changed citation 31 with the citation 39 (page 18, line 666).
|
|
Comments 7: Indicated in Fig. 6 F (table) the concentration of Berg, NAC and 4-PBA.
|
|
Response 7: Thanks a lot for your comments. The modification has been completed as requested (page 12, line 408).
|
|
Comments 8: Inhibition of noncp BVDV and the underlying mechanism needs to be investigated when conclusions on a clinical potential of Berg shall be drawn.
|
|
Response 8: Thanks a lot for your comments. In the manuscript, the antiviral effect and the corresponding mechanism of Berg against cytopathic strain NADL-1 were presented. Currently, the related experiments on the antiviral effect and mechanism of Berg against ncp BVDV strains are being carried out and have not yet been completed. The conclusion has been revised as suggested (page 1, line 25).
|
|
Comments 9: The authors claim a potential inhibition at the “release step”. This reviewer sees no valid data supporting that statement. The inhibition of apoptosis by Berg (line 483) is no valid argument since noncp BVDV is released from cells also in the absence of apoptosis.
|
|
Response 9: Thanks a lot for your comments. As suggested, we have revised the discussion in the manuscript and added the relevant references to support this idea. The modification has been marked in red (page 15, line 519-524). Increasing evidence showed that multiple viruses utilize apoptotic mimicry to facilitate their entrance and spread [49,50]. Moreover, some viruses have evolved to hijack apoptotic bodies at the late stage of infection for cell-cell transmission to evade host antiviral responses [45, 51]. Recent study has documented that BVDV activates ER stress-mediated apoptosis during late infection for virus proliferation and spread [35].
|
|
Comments 10: Line 59: “Flavivirus, the member of ER-tropic virus”; please change this sentence into better readable form. The cited paper deals with members of the genus “Flavivirus”; these viruses differ massively from members of the genus pestivirus. Thus, the citation seems not suited.
|
|
Response 10: Thanks a lot for your comments. We have re-written the paragraph to be more concise as suggested (page 2, line 65-75).
|
|
Comments 11: Line 439: “released stages” should read “release stage”.
|
|
Response 11: Thanks a lot for your comments. The modification has been completed as suggested (page 14, line 476).
|
|
Comments 12: Line 452: “Viruses interacted with” should read “Viruses interact with”.
|
|
Response 12: Thanks a lot for your comments. Changes were made as suggested (page 15, line 489).
|
|
4. Response to Comments on the Quality of English Language |
|
Point 1: Minor corrections required. |
|
Response 1: Thanks a lot for your suggestion, which is highly appreciated. We have carefully scrutinized the manuscript, and invited Prof. Guiyan Yang who graduated from University of California at Davis to help polish our manuscript. And we hope the revised manuscript could be acceptable for you. |

Round 2
Reviewer 2 Report
Comments and Suggestions for Authors
General Comments: The effects of Bergamottin (Berg), a natural furanocoumarin, on BVDV replication in MDBK cells is examined in this manuscript. The authors should be commended for having made substantial improvements to the manuscript. However, there are still some issues that need to be addressed. Minor grammatical edits are needed.
Specific Comments:
Methods:
1. The authors still need to state whether the MDBKs and fetal bovine serum were tested for noncytopathic BVDV. Many stocks of MDBKs and FBS are infected with adventitial ncp BVDV. Infection with ncp BVDV has important and confounding effects on the replication of other biotypes of BVDV.
2. The formula for cell viability doesn’t make sense. i.e., 1/1 = 1 or 100%. [A (experimental group) – A (blank group) / A (control group) – A (blank group)] × 100%.
3. There seems to be a temperature+time missing from the RTqPCR protocol.
Comments on the Quality of English LanguageThe quality of the English language is fine.
Author Response
|
Response to Reviewer 2 Comments
|
||
|
1. Summary |
|
|
|
Thank you very much for taking the time to review this manuscript. Please find the detailed responses below and the corresponding revisions in the re-submitted files.
|
||
|
2. Point-by-point response to Comments and Suggestions for Authors |
||
|
Methods: Comments 1: The authors still need to state whether the MDBKs and fetal bovine serum were tested for noncytopathic BVDV. Many stocks of MDBKs and FBS are infected with adventitial ncp BVDV. Infection with ncp BVDV has important and confounding effects on the replication of other biotypes of BVDV.
|
||
|
Response 1: Thanks a lot for your comments. In our experiments, we have tested fetal bovine serum for the presence of ncp BVDV using a commercial BVDV Ag/Serum plus test kit (99-43830, IDEXX Laboratories, Inc., Westbrook, ME, USA). This kit is applied for the detection of Erns antigen from serum samples. The results showed that fetal bovine serum was free of Erns antigen, indicating that it was not contaminated by BVDV (including cp and ncp biotype). Additionally, the total RNA was extracted from the MDBK cells and used for 5’UTR determination by RT-qPCR. The RT-qPCR analysis revealed that 5’UTR was not present in MDBK cells. Therefore, both the MDBK cells and fetal bovine serum were not contaminated by BVDV (including cp and ncp biotype). Furthermore, we have made a statement in our manuscript following your suggestion (page 3, line 101-102). |
||
|
Comments 2: The formula for cell viability doesn’t make sense. i.e., 1/1 = 1 or 100%. [A (experimental group) – A (blank group) / A (control group) – A (blank group)] × 100%. |
||
|
Response 2: We were really sorry for our careless mistakes. Thanks for your reminder. We have revised the text to address your concern (page 3, line 113-114). Cell viability (%) = [A (experimental group) – A (blank group)] / [A (control group) – A (blank group)] × 100.
|
||
|
Comments 3: There seems to be a temperature + time missing from the RT-qPCR protocol.
|
||
|
Response 3: Thanks a lot for your comments. As you suggested, we have added the detail of the RT-qPCR protocol in manuscript (page 3, line 126-130). The RT-qPCR assay was conducted on a 7500 Real-Time PCR System (Applied Biosystems; Thermo Fisher Scientific Inc.) using a qPCR Kit (AUQ-01, TranGen Biotechnology, Beijing, China). The RT-qPCR procedure was 95 °C for 30 s, and 45 amplification cycles in two steps of 95 °C for 5 s and 60 °C for 30 s. Then the melting curve was carried out as follows: 95 °C for 15 s, 60 °C for 60 s, 95 °C for 15 s and 60 °C for 15 s. |
||

Reviewer 3 Report
Comments and Suggestions for Authors
The authors responded satisfactorily
Author Response
Thanks a lot for your positive and constructive comments and suggestions on our manuscript.